# What Does Vision Tool-Use Reinforcement Learning Really Learn? Disentangling Tool-Induced and Intrinsic Effects for Crop-and-Zoom

**Yan Ma** [*3] **Weiyu Zhang** [*4] **Tianle Li** [5] **Linge Du** [2] **Xuyang Shen** [†2] **Pengfei Liu** [†16]

## Abstract

Vision tool-use reinforcement learning (RL) can equip vision–language models with visual operators such as crop-and-zoom and achieves strong performance gains, yet it remains unclear whether these gains are driven by improvements in tool use or evolving intrinsic capabilities. We introduce **MED** (Measure–Explain–Diagnose), a coarse-to-fine framework that disentangles intrinsic capability changes from tool-induced effects, decomposes the tool-induced performance difference into gain and harm terms, and probes the mechanisms driving their evolution. Across checkpoint-level analyses in the crop-and-zoom setting on two VLMs with different tool priors and six benchmarks, we find that improvements are dominated by intrinsic learning, while tool-use RL mainly reduces tool-induced harm (e.g., fewer call-induced errors and weaker tool schema interference) and yields limited progress in tool-based correction of intrinsic failures. Overall, in the crop-and-zoom setting studied here, current vision tool-use RL learns to coexist safely with tools rather than master them. Code: `https://github.com/GAIR-NLP/Med`

## 1. Introduction

Reinforcement learning (RL)–based post-training has become a central paradigm for improving large language models (e.g., DeepSeek-R1) (Guo et al., 2025), and recent efforts extend it to multimodal settings to boost visual and video reasoning performance (Li et al., 2025; Wang et al., 2025b; Yuan et al., 2025; Wang et al., 2025d). Yet, even strong

*Equal contribution [1]Shanghai Jiao Tong University [2]MiniMax [3]Fudan University [4]Peking University [5]The Chinese University of Hong Kong. Correspondence to: Xuyang Shen <shenxuyang@minimaxi.com>, Pengfei Liu <pengfei@sjtu.edu.cn>.

*Proceedings of the 43rd International Conference on Machine Learning*, Seoul, South Korea. PMLR 306, 2026. Copyright 2026 by the author(s).

vision-language models (VLMs) often treat the image as a static context: once an input is encoded, the model typically reasons without further visual interaction, relying on a single-pass perceptual snapshot. This creates a mismatch between how humans solve visually grounded problems and how current models do so. Humans routinely *interact* with visual evidence, zooming into regions, re-checking details, and verifying uncertain cues, especially when the correct decision hinges on fine-grained perception.

Vision tool-use RL aims to bridge this gap by equipping VLMs with explicit visual operators (e.g., crop-and-zoom) and training them to invoke tools during decoding (OpenAI, 2025; Zhang et al., 2025b; Zhou et al., 2025). Empirically, this paradigm yields sizeable gains on a wide range of multimodal benchmarks (OpenAI, 2025; Zhang et al., 2025b; Zhou et al., 2025), suggesting that interactive perception can complement intrinsic reasoning. A prominent instance is crop-and-zoom tool use (Su et al., 2025a; Zheng et al., 2025; Lai et al., 2025; Liu et al., 2025a), while other lines explore synthesizing and executing vision-related code (Zhang et al., 2025c; Zhao et al., 2025). However, a growing body of evidence also reveals a less flattering side: models may invoke tools redundantly or irrelevantly, and tool-use traces can be unfaithful or weakly grounded (Yu et al., 2025a; Liu et al., 2025b; Du et al., 2025). This has motivated work that regulates tool usage via reward shaping or call-level supervision, treating *rational* tool calling as a proxy for improved capability (Wang et al., 2025a;c; Liu et al., 2025b).

Despite this progress, a central question remains insufficiently understood: *what does vision tool-use RL actually learn?* Performance improvements observed under tool-available evaluation may arise from different sources. First, RL may strengthen the model's *intrinsic* capability, improving perception and reasoning even when tools are absent. Second, RL may improve *tool interaction itself*, including when-to-call decisions and execution quality. Third, RL may primarily reduce *tool-induced side effects*: lowering harms from tool availability (e.g., fewer harmful calls and less sensitivity to the tool schema), without materially improving the ability to correct tool-free failures. Existing evaluations typically report end-to-end tool-available accuracy, thereby hindering a mechanistic attribution of the gains.

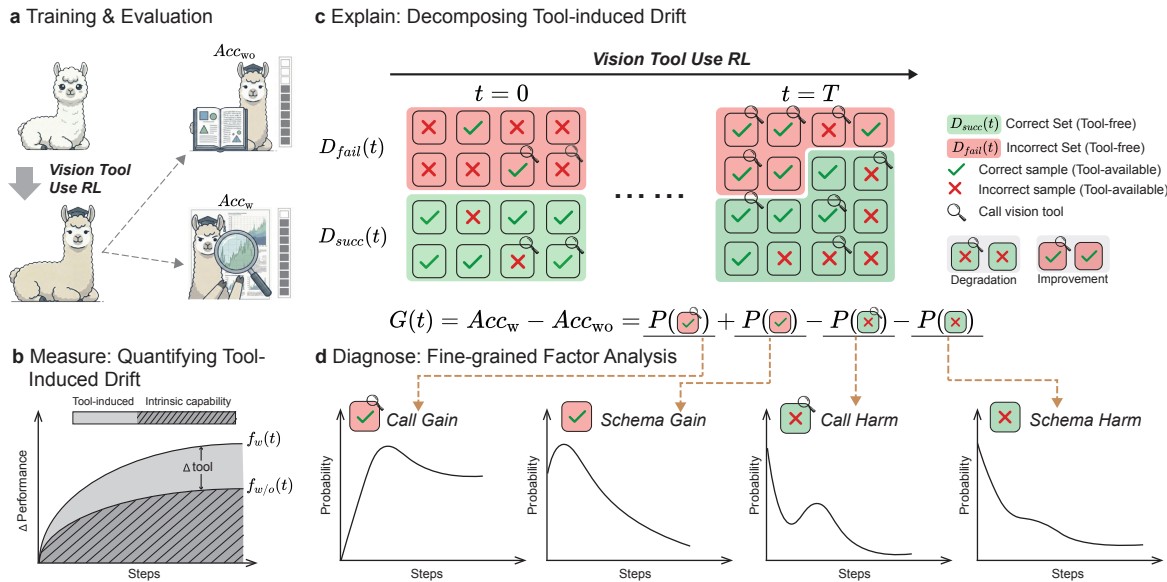

*Figure 1.* **The MED (Measure–Explain–Diagnose) framework for vision tool-use RL. (a)** We train a VLM with tool-use RL and evaluate each checkpoint under two protocols: *tool-free* accuracy $Acc_{wo}$ (tool-free performance) and *tool-available* accuracy $Acc_w$. **(b) Measure:** separate tool-free drift $f_{wo}(t) = Acc_{wo}(t) - Acc_{wo}(0)$ from tool-induced drift $\Delta_{\text{tool}}(t)$ by tracking the evolution of the gap $G(t) = Acc_w(t) - Acc_{wo}(t)$. **(c) Explain:** decompose $G(t)$ into Gains on tool-free failures $\mathcal{D}_{\text{fail}}(t)$ and Harms on tool-free successes $\mathcal{D}_{\text{succ}}(t)$, further distinguishing tool-call effects from no-call effects under tool availability (four terms to the right of the equal sign). **(d) Diagnose:** probe the underlying mechanisms behind each term's evolution to identify what changes in tool use drive gains or harms.

In this paper, we argue that answering the above question requires a training-dynamics view and attribution of performance changes. We therefore introduce a coarse-to-fine analysis framework, MED (Measure–Explain–Diagnose), designed to disentangle tool-free capability changes from tool-induced effects over RL training. MED first quantifies how much of the tool-available performance change can be explained by intrinsic improvement alone; it then decomposes the tool-induced performance difference into interpretable gain and harm components; finally, it diagnoses the underlying mechanisms behind these components to distinguish changes in tool-use potential, calling behavior, and tool-use quality. This analysis is complementary to call-level faithfulness studies (Yu et al., 2025a; Liu et al., 2025b; Du et al., 2025): rather than asking only whether a tool call looks reasonable, we ask *why* tool availability helps or hurts, and *which* learning signals dominate the observed improvements.

We instantiate our study in a widely used minimal setting, *crop-and-zoom* (Su et al., 2025a; Zheng et al., 2025; Liu et al., 2025a), and conduct checkpoint-level analyses across two representative VLM backbones and six standard benchmarks (Lai et al., 2025; Wu & Xie, 2023; Wang et al., 2025e). Importantly, the two backbones differ not only in strength but also in prior exposure to the target tool: one is tool-naive (not explicitly trained on crop-and-zoom), whereas the other is tool-native (trained with this tool upon release), en-

abling us to probe whether the same training dynamics hold across distinct tool-familiarity regimes (Bai et al., 2025b;a). Our empirical findings are therefore scoped to this crop-and-zoom setting, while MED only requires paired tool-free/tool-available evaluation and observable call/no-call behavior. Together, our results provide a mechanistic and attribution-grounded answer to the titular question within this setting.

Our work makes three contributions: 1) We propose MED (Measure–Explain–Diagnose), a coarse-to-fine framework that disentangles tool-induced effects from intrinsic capability drift in vision tool-use RL. 2) We derive an interpretable decomposition of the tool-induced performance gap into Gross Gain/Harm and further factorize each effect to enable mechanism-level diagnosis of when vision tools help, when they hurt, and why. 3) Through extensive checkpoint-level analyses on two VLMs and six benchmarks in the crop-and-zoom setting, we find that vision tool-use RL mainly reduces tool-induced harm (lower breakage on intrinsic successes) but shows limited improvement in tool-based correction of intrinsic failures.

## 2. Related Work

Recent RL–based post-training paradigms (e.g., DeepSeek-R1 (Guo et al., 2025)) have been extended to multimodal LLMs (Li et al., 2025; Wang et al., 2025b; Yuan et al., 2025;

Wang et al., 2025d), yielding broad gains on visual and video reasoning benchmarks. However, these methods typically optimize textual reasoning trajectories without explicit mechanisms to *adaptively interrogate* visual inputs during inference (e.g., querying regions, verifying evidence, or refining observations), limiting interactive visual reasoning.

To enable such interaction, *vision tool-use RL* (OpenAI, 2025) equips models with visual operators and trains them to invoke tools during decoding (OpenAI, 2025; Zhang et al., 2025b; Zhou et al., 2025). A prominent instance is crop-and-zoom tool use (Su et al., 2025a; Zheng et al., 2025; Lai et al., 2025; Liu et al., 2025a; Zhang et al., 2025a; 2026), while other lines explore synthesizing and executing vision-related code (Zhang et al., 2025c; Zhao et al., 2025). These approaches report strong benchmark improvements, but emerging evidence suggests that models may call tools redundantly or irrelevantly, producing unfaithful or ungrounded tool-use traces (Yu et al., 2025a; Liu et al., 2025b; Du et al., 2025).

To improve the faithfulness of tool-use, several works regulate visual tool usage by augmenting rewards with signals such as utility estimation (Wang et al., 2025a), alignment with human rationales (Wang et al., 2025c), or causal relevance (Liu et al., 2025b). These studies primarily operate at the level of *call appropriateness*, asking whether a tool invocation is necessary or aligned with human reasoning. In contrast, fewer works explicitly attribute *overall performance changes* in tool-use RL to intrinsic capability drift versus tool-induced effects, or analyze the mechanisms through which gains and harms evolve during training. Moreover, existing analyses are often conducted under a single backbone and a single tool-familiarity regime. In our study, we analyze training dynamics across two representative VLMs that differ not only in backbone strength but also in prior exposure to crop-and-zoom: Qwen2.5VL has not been explicitly trained with this tool, whereas Qwen3VL has been.

## 3. Methodology

**Preliminaries and Problem Formulation** We study a VLM, denoted as $\Phi$, trained with RL to solve a set of visual tasks. We focus on the *Vision Tool-Use RL* setting, where $\Phi$ may invoke visual tools (e.g., crop-and-zoom) during decoding to assist prediction.

To track learning over training time $t \in [0, T]$, we evaluate each checkpoint under two inference protocols (Fig. 1a):

**Tool-free:** The model is not given any external tools and answers from the original visual input. Let $Acc_{wo}(t)$ denote the task accuracy at checkpoint $t$ under this tool-free protocol, which we use as a practical reference for tool-free behavior.

**Tool-available:** The model is provided with the tool schema and may invoke the tool during decoding. Let $Acc_w(t)$ denote the task accuracy at checkpoint $t$ under this protocol. We use *intrinsic capability* to refer to the model's ability to solve a task without executing external tools. Operationally, we measure it using the tool-free protocol, i.e., $Acc_{wo}(t)$ on the same checkpoint. This is a practical operationalization rather than a claim that all tool-independent capability is perfectly isolated from tool exposure during RL training. Both quantities are measured on the same checkpoint; the contrast is between inference protocols, not between separately trained models.

We measure progress as the change in accuracy from the initial checkpoint ($t = 0$), and define the tool-free and tool-available drifts as

$$f_{wo}(t) = Acc_{wo}(t) - Acc_{wo}(0), \ f_w(t) = Acc_w(t) - Acc_w(0) \tag{1}$$

Here, $f_{wo}(t)$ measures the change in tool-free accuracy, whereas $f_w(t)$ measures the end-to-end accuracy change when tool use is available. Our goal is to decompose $f_w(t)$ into a tool-free component, captured by $f_{wo}(t)$ under the paired protocol above, and a tool-induced component. Specifically, we test whether gains in $Acc_w(t)$ arise from improved tool interaction (e.g., when/how to call) or from improvements already reflected under tool-free inference. To this end, we develop a coarse-to-fine analysis framework, termed MED (Measure-Explain-Diagnose), to attribute performance changes to tool-free drift versus tool-induced effects.

### 3.1. Measure: Quantifying Tool-Induced Drift

We define *tool-induced performance gap* at checkpoint $t$:

$$G(t) \triangleq Acc_w(t) - Acc_{wo}(t) \tag{2}$$

$G(t)$ represents the instantaneous performance difference induced by tool access relative to the tool-free protocol. We then track the evolution of this performance gap over training by defining $\Delta_{tool}(t) \triangleq G(t) - G(0)$, which captures the *tool-induced drift*—the performance shift resulting from the model's evolving use of the tool. Using Eq. (1), we obtain an additive decomposition of the tool-available drift:

$$\underbrace{f_w(t)}_{\text{Tool-available drift}} = \underbrace{f_{wo}(t)}_{\text{Tool-free drift}} + \underbrace{\Delta_{tool}(t)}_{\text{Tool-induced drift}} \tag{3}$$

Eq. (3) expresses the tool-available drift as the sum of the tool-free drift measured under the paired tool-free protocol and the change in tool-induced performance gap over training.

To summarize contributions over the training horizon $t \in [0, T]$, we measure the cumulative *magnitude* of each drift

component by integrating their absolute values (Fig. 1b):

$$|B_{\text{wo}}| \triangleq \int_0^T |f_{\text{wo}}(t)| \, dt, \quad |B_{\Delta\text{tool}}| \triangleq \int_0^T |\Delta_{\text{tool}}(t)| \, dt \tag{4}$$

Finally, we define the *tool contribution ratio* as the fraction of total drift magnitude attributed to tool-induced drift:

$$S_{\text{tool}} = \frac{|B_{\Delta\text{tool}}|}{|B_{\text{wo}}| + |B_{\Delta\text{tool}}|} \tag{5}$$

When $|B_{\text{wo}}| + |B_{\Delta\text{tool}}|$ is non-negligible, $S_{\text{tool}} \approx 0$ indicates dominance by tool-free drift magnitude, whereas $S_{\text{tool}} \approx 1$ indicates dominance by the magnitude of tool-induced drift. Fig. 1b illustrates these quantities as the total area traced by $f_{\text{wo}}(t)$ (shaded in dark grey, representing tool-free drift) and the area between $f_{\text{w}}(t)$ and $f_{\text{wo}}(t)$ (filled in light grey, representing the tool-induced drift) over time.

### 3.2. Explain: Decomposing Tool-induced Drift

While $S_{\text{tool}}$ quantifies the overall tool-induced drift, it reflects only the magnitude of this drift relative to the total drift. However, it does not explain the underlying dynamics, such as how tool-induced effects (gains and harms) evolve over training. To gain a deeper understanding of training dynamics, we need to decompose $\Delta_{\text{tool}}(t) = G(t) - G(0)$ further. Since $G(t)$ measures the performance gap between $Acc_{\text{w}}(t)$ and $Acc_{\text{wo}}(t)$, we decompose it by asking which examples change when moving from the tool-free protocol to the tool-available protocol.

**Two-step accounting.** First, the tool-free protocol at checkpoint $t$ partitions the task set $\Omega$ into two disjoint subsets: the *failure set* $\mathcal{D}_{\text{fail}}(t)$, where the model fails without tools, and the *success set* $\mathcal{D}_{\text{succ}}(t)$, where it succeeds without tools. Second, under the tool-available protocol, each example either calls or does not call the tool, and either becomes correct or wrong. This yields gains when tool-free failures are recovered and harms when tool-free successes are lost.

**Probabilistic Decomposition of $G(t)$.** We analyze the tool available accuracy $Acc_{\text{w}}(t)$ by conditioning on tool usage. Let $c$ denote the event of calling the tool, and $\checkmark$ (or $\times$) denote a correct (or incorrect) prediction under the tool-available protocol. By the law of total probability:

$$Acc_{\text{w}}(t) = P(\checkmark \mid \mathcal{D}_{\text{fail}})P(\mathcal{D}_{\text{fail}}) + P(\checkmark \mid \mathcal{D}_{\text{succ}})P(\mathcal{D}_{\text{succ}}) \tag{6}$$

Moreover, conditioning on tool usage gives, for any subset $\mathcal{D} \subseteq \Omega$, $P(\checkmark \mid \mathcal{D}) = P(c \mid \mathcal{D})P(\checkmark \mid c, \mathcal{D}) + P(\neg c \mid \mathcal{D})P(\checkmark \mid \neg c, \mathcal{D})$. Subtracting the tool-free reference $Acc_{\text{wo}}(t) = P(\mathcal{D}_{\text{succ}})$ from Eq. (6), we rewrite the success-side term via $P(\checkmark \mid \mathcal{D}_{\text{succ}}) = 1 - P(\times \mid \mathcal{D}_{\text{succ}})$, and further expand $P(\times \mid \mathcal{D}_{\text{succ}})$ by $c/\neg c$ analogously. This yields the four-term decomposition of the tool-induced gap

$G(t)$ as an accounting identity (see illustration in Fig. 1c and full derivation in §A):

$$
\begin{aligned}
G(t) = &\underbrace{P(\mathcal{D}_{\text{fail}})P(c \mid \mathcal{D}_{\text{fail}})P(\checkmark \mid c, \mathcal{D}_{\text{fail}})}_{\text{Term 1: Call Gain}} \\
&+ \underbrace{P(\mathcal{D}_{\text{fail}})P(\neg c \mid \mathcal{D}_{\text{fail}})P(\checkmark \mid \neg c, \mathcal{D}_{\text{fail}})}_{\text{Term 2: Schema Gain}} \\
&- \underbrace{P(\mathcal{D}_{\text{succ}})P(c \mid \mathcal{D}_{\text{succ}})P(\times \mid c, \mathcal{D}_{\text{succ}})}_{\text{Term 3: Call Harm}} \\
&- \underbrace{P(\mathcal{D}_{\text{succ}})P(\neg c \mid \mathcal{D}_{\text{succ}})P(\times \mid \neg c, \mathcal{D}_{\text{succ}})}_{\text{Term 4: Schema Harm}}
\end{aligned}
\tag{7}
$$

**Term Interpretation.** Each term represents a specific change relative to the tool-free reference:

**Term 1 (Call Gain):** Tool-free failures *corrected* after calling the tool. This asks whether tool execution recovers examples that were wrong without tools.

**Term 2 (Schema Gain):** Tool-free failures *recovered* under the tool-available protocol without invocation. We interpret this as schema-conditioned no-call recovery, not as a pure causal effect of the schema itself.

**Term 3 (Call Harm):** Tool-free successes *lost* after tool calls. This asks whether invoking the tool breaks examples that were already solvable without tools.

**Term 4 (Schema Harm):** Tool-free successes *lost* under the tool-available protocol without invocation. This captures no-call errors under tool availability, including possible schema interference.

Eq. (7) makes the changes in $G(t)$ actionable rather than only aggregate. By isolating *Gross Gain* (Terms 1+2) from *Gross Harm* (Terms 3+4), and separating call-based effects (Terms 1 and 3) from no-call effects under tool availability (Terms 2 and 4), we can ask whether a method improves tool-based recovery, reduces tool-induced breakage, or mainly changes no-call behavior. Unlike $S_{\text{tool}}$, which provides a *quantitative* measure of the drift's magnitude, this decomposition explains which type of gain or harm drives its evolution. By resolving $G(t)$ into its components, we can identify the specific drivers of $\Delta_{\text{tool}}(t)$, determining whether the observed drift is dominated by emerging utility (*Gain-dominant*) or suppressed by interference (*Harm-dominant*). This explains how the tool impacts learning dynamics beyond the aggregate statistics.

### 3.3. Diagnose: Fine-grained Factor Analysis

While Eq. (7) tells us which type of gain or harm changes, each term is the product of three probabilities. The Mass–Policy–Quality factorization then asks why a given term

changes over training. For instance, a decline in Call Gain could result from three mechanisms: a shrinking failure set ($P(\mathcal{D}_{\text{fail}}) \downarrow$), a lower calling probability ($P(c \mid \mathcal{D}_{\text{fail}}) \downarrow$), or degraded execution ($P(\checkmark \mid c, \mathcal{D}_{\text{fail}}) \downarrow$).

To pinpoint the root cause, we decompose the problem into finer components. Each term is the product of three variables: *Mass*, *Policy*, and *Quality*.

**Factor Definition.** For each term in Eq. (7), defined by a tuple of domain $\mathcal{D} \in \{\mathcal{D}_{\text{fail}}, \mathcal{D}_{\text{succ}}\}$, action $a \in \{c, \neg c\}$, and outcome $o \in \{\checkmark, \times\}$, we decompose it as:

$$\text{Term}(\mathcal{D}, a, o) = \underbrace{P(\mathcal{D})}_{\text{Mass}} \cdot \underbrace{P(a \mid \mathcal{D})}_{\text{Policy}} \cdot \underbrace{P(o \mid a, \mathcal{D})}_{\text{Quality}} \quad (8)$$

*Mass (M)*: The size of the domain (e.g., $P(\mathcal{D}_{\text{fail}})$). This represents the *capacity* available for the tool to generate gain or harm. *Policy ($\pi$)*: The conditional probability of taking action $a$ (e.g., $P(c \mid \mathcal{D}_{\text{fail}})$). This reflects the model's *decision-making strategy* ("When to call"). *Quality (Q)*: The conditional probability of outcome $o$ given action $a$ and domain $\mathcal{D}$ (e.g., $P(\checkmark \mid c, \mathcal{D}_{\text{fail}})$), reflecting the model's *execution capability* ("How to use") within that domain.

**Diagnostic Insights.** Eq. (8) uncovers two critical training dynamics hidden by aggregate metrics. 1) The Intrinsic-Tool Trade-off (Mass Dynamics): As the model's intrinsic capability improves, the failure set $\mathcal{D}_{\text{fail}}$ shrinks. This reduction in Mass limits the upper bound of Call Gain, potentially causing $G(t)$ to plateau even if the tool execution quality ($Q$) improves. 2) Policy-Quality Decoupling: We can distinguish between learning *to attempt* and learning *to succeed*. An increase in tool usage (Policy) without accuracy improvement (Quality) may indicate a bias to invoke tools rather than true tool proficiency.

**Implementation.** We estimate these probabilistic factors using empirical frequencies over evaluation benchmarks at each checkpoint. This enables us to trace the *temporal evolution* of Mass, Policy, and Quality, visualizing the underlying learning dynamics across different tasks.

## 4. Experiment

### 4.1. Experimental Setup

We summarize our setup below. See §B for details of the visual tool, models, data, decontamination, training, inference and evaluation.

**Vision Tool.** We study vision tool-use RL with a single visual tool, *Crop-and-Zoom*, which extracts high-resolution details from image regions. The tool schema is injected into the system prompt during training and inference.

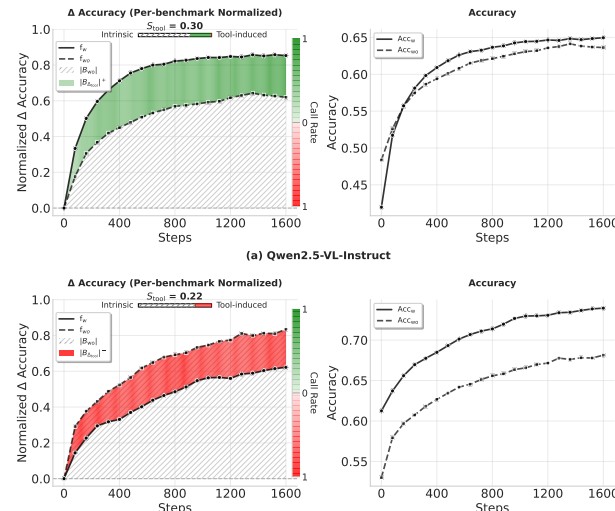

*Figure 2.* **Quantifying Intrinsic and Tool-Induced Drift.** We aggregate learning dynamics across six benchmarks (VStar, HR-Bench 4k/8k, VisualProbe Easy/Medium/Hard), evaluated every 80 gradient steps (21 checkpoints). **Left:** Tool-free and tool-available drifts. We normalize per-benchmark $\Delta$ accuracy to $[-1, 1]$ and then average across benchmarks to compute curves. The grey area ($|B_{\text{wo}}|$) quantifies the cumulative magnitude of intrinsic drift ($f_{\text{wo}}$). The colored area represents the magnitude of tool-induced drift ($\Delta_{\text{tool}}$). Green indicates positive relative gain ($f_w > f_{\text{wo}}$), while red indicates negative relative drift ($f_w < f_{\text{wo}}$). *Color intensity* corresponds to the tool call rate. The *top progress bar* displays the tool contribution ratio ($S_{tool}$), i.e., the proportion of total drift magnitude attributed to tool effects. **Right:** Absolute accuracy ($Acc_w$ and $Acc_{wo}$) is averaged directly across all benchmarks. All curves are smoothed for visualization only. Full details on area calculation, normalization, aggregation and smoothing are provided in §D.

**Models.** We experiment with two VLMs, Qwen2.5-VL-Instruct-7B and Qwen3-VL-Instruct-8B. Both models support function calling but differ in prior tool familiarity: Qwen2.5-VL has not been explicitly trained with Crop-and-Zoom, whereas Qwen3-VL has been trained with it. During all RL experiments, we fine-tune the LLM backbone while keeping the vision encoder frozen.

**Data.** RL training is conducted on a composite dataset of approximately 15k samples, each consisting of an image, a question, and a verifiable answer. The dataset consists primarily of high-resolution VQA from Thyme (Zhang et al., 2025c) and Mini-O3 (Lai et al., 2025) ($\sim$80%), to encourage active tool usage, and is complemented by a diverse long tail ($\sim$20%) to improve robustness across domains. We apply visual decontamination by excluding training images with pHash Hamming distance $< 5$ from any evaluation image.

**Training and Evaluation.** We train models with Group Relative Policy Optimization (GRPO) and a binary outcome-based reward defined solely by final-answer correctness,

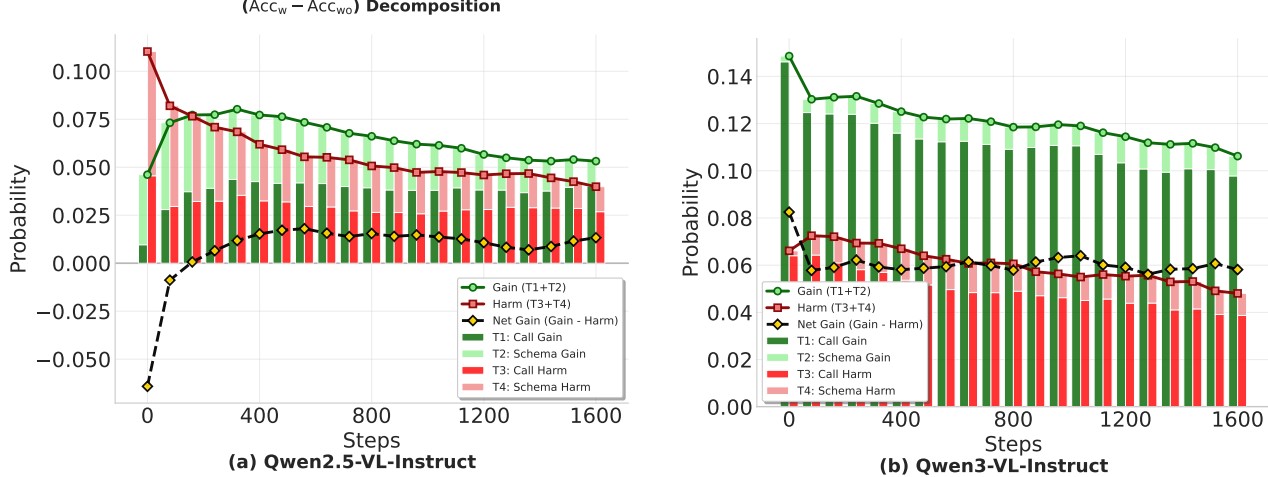

*Figure 3.* **Decomposition of Tool-Induced Performance Gap** $G(t)$**.** Averaged across six benchmarks. Eq. (7) breaks down the net gap $G(t)$ (yellow diamonds) into *Gross Gain* (green; T1+T2) and *Gross Harm* (red; T3+T4). *Gross Gain* consists of Call Gain (T1; tool-free failures corrected via tool execution) and Schema Gain (T2; schema-conditioned no-call recovery under tool availability). *Gross Harm* consists of Call Harm (T3; tool-free successes flipped to errors after tool calls) and Schema Harm (no-call errors under tool availability, including possible schema interference). (a) Qwen2.5-VL: Call Gain (T1) quickly reverses the initial negative gap, then plateaus and slightly declines. (b) Qwen3-VL: Gross Gain shrinks mainly due to declining Call Gain. Both models show concurrent reductions in Gross Gain and Gross Harm at later stages. Overall, the dynamics are consistent with harm reduction (suppressed detrimental usage) rather than continued gain maximization.

without tool reward or curriculum learning. We evaluate models under both tool-available and tool-free protocols on six benchmarks (VStar, HR-Bench 4k/8k, VisualProbe Easy/Medium/Harm), with greedy decoding (temperature = 0) at regular checkpoints throughout training.

### 4.2. [Exp-I] Measure: The Dominance of Intrinsic Drift

We begin by applying the *Measure* component to quantify intrinsic versus tool-induced sources of performance change. Fig. 2 illustrates the evolution of intrinsic drift, operationalized by tool-free accuracy change ($f_{wo}$), and tool-available drift ($f_w$), aggregated across all six evaluation benchmarks. There are **three key observations**:

**Intrinsic drift dominates overall performance change.** Contrary to the common intuition that vision tool-use RL primarily optimizes tool handling, we find that most performance gains are driven by intrinsic improvement as measured by our tool-free protocol. The grey area ($B_{wo}$) dominates the drift curves for both models. The tool contribution ratio $S_{tool}$ remains low (0.30 for Qwen2.5-VL and 0.22 for Qwen3-VL), indicating that over 70% of learning progress is captured by the intrinsic component operationalized by tool-free inference. This suggests that, in this crop-and-zoom setting, vision tool-use RL largely improves general capability while the additional tool-induced component remains comparatively smaller.

**Relative drift diverges across initialization regimes.** We observe a distinct divergence in how tool-induced drift evolves between two models. For Qwen2.5VL (no prior crop-and-zoom training), the tool-available drift exceeds intrinsic drift ($f_w > f_{wo}$), resulting in a *positive* gain (green area). In contrast, Qwen3VL (has prior crop-and-zoom training) exhibits a *negative* relative drift ($f_{wo} > f_w$). While absolute accuracy continues to improve, intrinsic capability improves *faster* than tool-assisted performance.

**Absolute performance improves monotonically.** The right column of Fig. 2 provides a sanity check. Despite the negative relative drift in Qwen3-VL, the absolute accuracy for both $Acc_w$ (solid line) and $Acc_{wo}$ (dashed line) increases monotonically. The right plots highlight the initialization gap: Qwen3-VL starts with a significantly higher baseline and a larger initial tool gap ($Acc_w > Acc_{wo}$), whereas Qwen2.5-VL starts lower with a negligible gap. Thus, the "red area" in Qwen3-VL does not imply forgetting of tool skills, but rather a shift in reliance: the tool becomes less critical as intrinsic capabilities expand.

### 4.3. [Exp-II] Explain: Mitigating Harm Rather Than Maximizing Gain

In §4.2, we found that tool-induced effects account for only a small fraction of overall drift. To explain this constraint, we decompose the tool-induced gap $G(t)$ (Eq. (7)) into *Gross Gain* (green; T1+T2) and *Gross Harm* (red; T3+T4). Fig. 3 shows two concurrent trends: Gross Gain stagnates

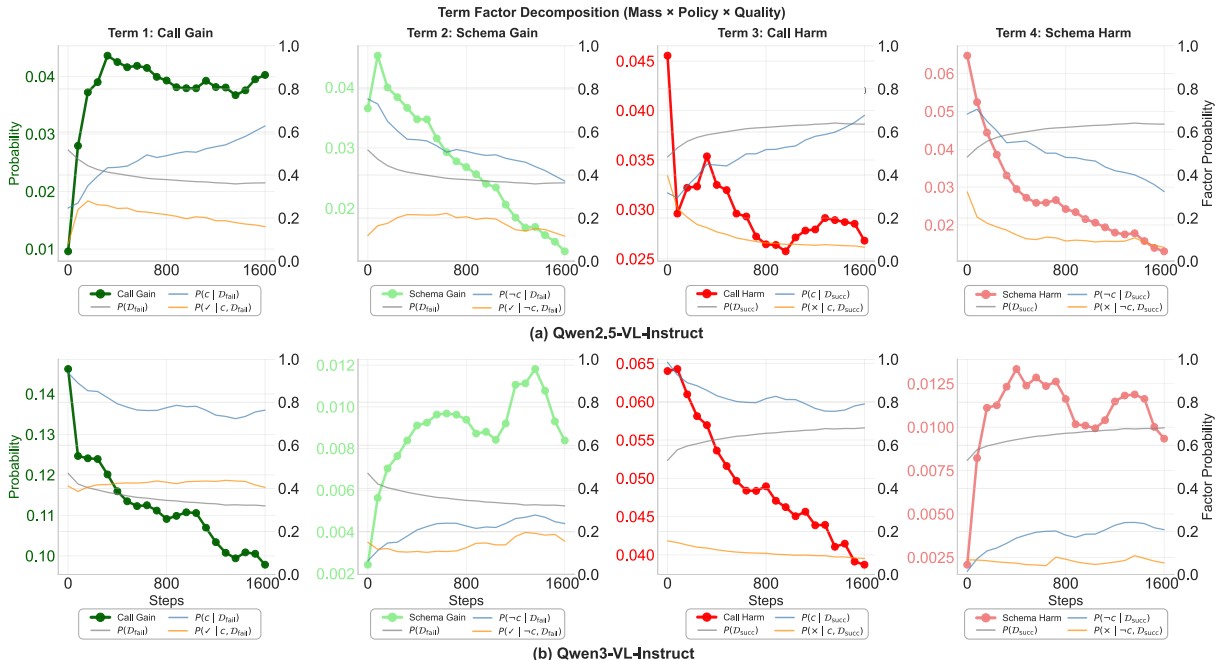

*Figure 4.* **Factor Decomposition of Tool-Induced Effects.** We show the temporal evolution of the four terms, factorized into Mass, Policy, and Quality, for (a) Qwen2.5-VL-Instruct and (b) Qwen3-VL-Instruct. In each subplot, the thick line shows the term value (left axis), and thin lines show its factors (right axis): *Mass* (grey, $P(\mathcal{D})$), *Policy* (blue, $P(a \mid \mathcal{D})$), and *Quality* (orange, $P(o \mid a, \mathcal{D})$).

while Gross Harm decreases, jointly limiting the net gap.

**The Stagnation of Gross Gain.** The main constraint on further widening $G(t)$ is the trajectory of *Gross Gain* (green bars). Specifically, Term 1 (Call Gain)—intrinsic failures corrected via tool calls—does not keep increasing. Instead of steadily increasing, Term 1 plateaus after an early rise (Qwen2.5-VL) or declines monotonically (Qwen3-VL). This indicates saturation in the model's ability to extract additional tool-based gains.

**The Consistent Reduction of Gross Harm.** In contrast, Gross Harm (red bars) decreases consistently across both models. We observe a steady decline in tool-induced penalties: Qwen2.5-VL mainly reduces Schema Harm (T4), whereas Qwen3-VL mainly reduces Call Harm (T3). This points to improved robustness to schema interference versus fewer harmful calls. In both cases, the aggregate effect is a continued reduction in tool-induced harm. This indicates effective interference management: tools become progressively less detrimental to intrinsic inference.

**The counterbalancing of Gain and Harm.** Combining these observations explains the stagnation of the net performance gap $G(t)$ (black curve in Fig. 3). This plateau reflects a counterbalance: reduced Gross Harm is offset by saturated/declining Gross Gain. In short, harm decreases but gain does not increase, keeping $G(t)$ roughly constant. However,

this decomposition represents the *outcome*, not the *root cause*. For Call Gain (T1), is the stagnation driven by reduced calling policy or lower execution success? To decouple policy ($P(c \mid \mathcal{D})$) from execution quality ($P(\checkmark \mid c, \mathcal{D})$), we factorize each term into its probability components in the next section.

### 4.4. [Exp-III] Diagnose: Suppressing Conditional Errors Rather Than Enhancing Correction

To pinpoint the root causes of Exp-II, we factorize each term into Mass ($P(\mathcal{D})$), Policy ($P(a \mid \mathcal{D})$), and Quality ($P(o \mid a, \mathcal{D})$), as defined in Eq. (8). Fig. 4 traces the temporal evolution of these factors.

**Limited failure correction, but reduced breakage on successes.** We analyze execution quality in Fig. 4. For Call Gain (T1), the correction success rate on intrinsic failures, $P(\checkmark \mid c, \mathcal{D}_{\text{fail}}(t))$, is flat for Qwen3-VL and rises early then declines for Qwen2.5-VL. Because $\mathcal{D}_{\text{fail}}$ shrinks and becomes increasingly difficult over training, this trend must be interpreted with care; we control for this moving-target effect in §4.5. In contrast, for Call Harm (T3), the breakage rate on intrinsic successes, $P(\times \mid c, \mathcal{D}_{\text{succ}}(t))$, decreases consistently. Overall, RL primarily suppresses tool-induced errors on already-solved instances rather than strengthening tool-based failure correction.

*Table 1.* **Manual taxonomy of persistent failures.** We label persistent tool-free failures on Qwen2.5-VL that still fail under tool-available evaluation.

| Failure type | $n$ | Breakdown |
|---|---|---|
| No call, wrong | 100 | should have called: 82; not clearly a missed call: 18 |
| Call, wrong | 99 | crop wrong: 52; crop right but answer wrong: 37; crop right but still hard: 10 |

**Case-level analysis of persistent failures.** To better understand why hard-case correction remains limited, we manually inspect persistent tool-free failures, $\mathcal{D}_{\text{fail}}(0) \cap \mathcal{D}_{\text{fail}}(t_{\text{final}})$, on Qwen2.5-VL at the final checkpoint, focusing on VStar, HR-Bench 4K, and VisualProbe Hard. Within this cohort, 370 cases still fail under tool-available evaluation: 271 no-call failures and 99 call-but-still-wrong failures. Two authors label all 99 called failures and a random sample of 100 no-call failures.

This descriptive taxonomy shows that persistent failures are not dominated by a single error mode: many are missed opportunities to call, while called failures often come from wrong crops or weak use of cropped evidence. In "crop right but answer wrong," the crop contains sufficient evidence under human inspection but the model still answers incorrectly; in "crop right but still hard," the crop localizes the correct region but the target remains too small, ambiguous, or visually difficult after zooming.

**RL Mitigates the Interference of Tool Schema.** Beyond execution quality, training suppresses schema-induced effects. The schema effect depends on prior exposure: Qwen3VL shows minimal Schema Gain/Harm (both $\approx$ 0.01). Conversely, Qwen2.5VL is initially sensitive to schema effects. However, over the course of training, RL drives down both Schema Gain and Schema Harm. This suggests a common optimization direction: reducing sensitivity to the tool schema so that its presence does not distract intrinsic inference.

### 4.5. Sanity Checks and Analysis

**Justification of the Tool-free Intrinsic Reference.** Following the operational definition above, $Acc_{\text{wo}}$ measures intrinsic capability through tool-free inference on the same checkpoint. While one might argue for using $Acc_{\text{schema}}$ (where the schema is provided but execution is forbidden) as the reference to maintain prompt consistency with RL training, this setting changes the inference protocol by showing the tool schema while artificially preventing execution. In Tab. 3, forcing this schema-shown-but-execution-forbidden protocol substantially reduces accuracy (5.8% for Qwen2.5VL and 13.0% for Qwen3VL), suggesting that it is not a better reference setting. Using $Acc_{\text{schema}}$ as the base-

*Table 2.* **No-tool RL check for the tool-free reference.** Both settings use the same 200-step RL pipeline; no-tool RL removes tool access during RL training. Mean gap is the mean absolute per-checkpoint difference in $Acc_{\text{wo}}$ across 21 checkpoints.

| Backbone | No-tool RL final $Acc_{\text{wo}}$ | Tool-RL final $Acc_{\text{wo}}$ | Mean gap $|\Delta Acc_{\text{wo}}|$ |
|---|---|---|---|
| Qwen2.5-VL | 0.644 | 0.636 | 0.0175 |
| Qwen3-VL | 0.678 | 0.681 | 0.0191 |

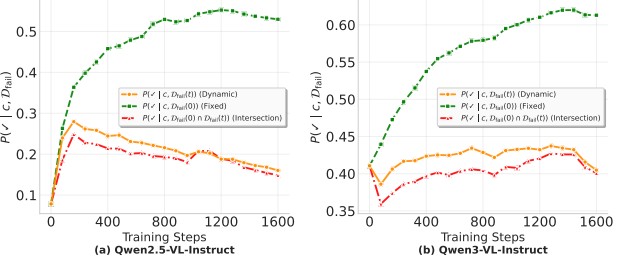

(a) Qwen2.5-VL-Instruct     (b) Qwen3-VL-Instruct

*Figure 5.* **Robustness to the Moving Failure Set.** The Call-Gain quality $P(\checkmark \mid c, \mathcal{D}_{\text{fail}})$ evaluated under different failure-set definitions: the current failure set $\mathcal{D}_{\text{fail}}(t)$ (Dynamic), the fixed initial cohort $\mathcal{D}_{\text{fail}}(0)$ (Fixed), and persistent failures $\mathcal{D}_{\text{fail}}(0) \cap \mathcal{D}_{\text{fail}}(t)$. Improvement is observed on the fixed cohort but remains limited on the current and persistent failure sets.

line would therefore overstate tool gains by treating recovery from this artificial constraint as improvement. We further check $Acc_{\text{wo}}$ from the training side by running no-tool RL with the same 200-step pipeline but without tool access during RL training. As shown in Tab. 2, no-tool RL closely tracks the tool-free accuracy of outcome-only tool-RL on both backbones. This supports using $Acc_{\text{wo}}$ as a practical intrinsic reference, while not claiming perfect isolation of all tool-independent capability.

**Alignment of Tool Call Gains with Human Logic.** While the metric $P(\checkmark \mid c, \mathcal{D}_{\text{fail}})$ (in Term 1) captures the contribution of tool calls, it does not reveal whether the underlying mechanism aligns with human logic. A statistical gain could arise from explicit alignment (e.g., cropping the target as a human would) or from implicit shortcuts (e.g., latent distribution shifts irrelevant to the visual content). We investigate whether the learned tool-use behavior is interpretable and aligned with human expectations.

To quantify this, we collect all "Call Gain" samples ($N = 269$) from the evaluation results of all benchmarks at the final checkpoint. Two PhD students and Gemini-3-Pro independently evaluate each sample using the multi-turn response and visual context (see the prompt in §F). We report the intersection (Human $\cap$ AI) of their positive judgments as human-aligned gains to ensure reliability. In Tab. 4, both models demonstrate greate alignment with human ($> 60\%$). Specifically, Qwen3-VL, having prior training to this tool, achieves near-perfect interpretability (93.0%),

whereas Qwen2.5-VL, with no prior tool tuning, exhibits a bit of shortcut-driven behaviors. This verification confirms that the *quality* of the gains is largely aligned with human.

**Robustness to the Moving Failure Set.** Because $\mathcal{D}_{\text{fail}}(t)$ is defined by the tool-free model at checkpoint $t$, it shrinks and shifts in difficulty over training. As a result, $P(\checkmark \mid c, \mathcal{D}_{\text{fail}})$ alone may conflate improvements in execution quality with the increasing difficulty of the failure set over training. To control for this, we additionally evaluate $P(\checkmark \mid c, \cdot)$ on (i) a fixed initial failure cohort $\mathcal{D}_{\text{fail}}(0)$ and (ii) persistent failures $\mathcal{D}_{\text{fail}}(0) \cap \mathcal{D}_{\text{fail}}(t)$ (fail at initialization and remain unsolved without tools at checkpoint $t$). In Fig. 5, we find that $P(\checkmark \mid c, \mathcal{D}_{\text{fail}}(0))$ increases substantially, whereas the current and persistent-failure estimates show little improvement, suggesting that quality gains do not extend to the remaining hardest failures.

### 4.6. What Does Vision Tool-Use RL Really Learn?

Synthesizing Exp-I–III and sanity checks, we answer the central question. Contrary to the ideal of tool mastery, in this crop-and-zoom setting, current vision tool-use RL learns a more conservative policy:

1. **Limited Contribution:** Tool-induced effects remain a minor component of overall improvement. While tool access contributes to performance, its effect is limited compared to intrinsic improvements.

2. **Interference Management:** The model reduces Gross Harm by suppressing execution errors ($P(\times \mid c, \mathcal{D}_{\text{succ}}) \downarrow$) and mitigating schema distraction.

3. **Limited Failure Correction on Hard Cases:** Call-Gain quality $P(\checkmark \mid c, \mathcal{D}_{\text{fail}})$ shows little improvement on the current failure set and on persistent failures $\mathcal{D}_{\text{fail}}(0) \cap \mathcal{D}_{\text{fail}}(t)$, indicating no strengthening of tool-based correction on instances that remain unsolved without tools, even though $P(\checkmark \mid c, \mathcal{D}_{\text{fail}}(0))$ can increase on the fixed initial-failure cohort.

**Training variants and algorithmic implication.** Since our main experiments use outcome-only rewards, we further compare no-tool RL and a tool-aware reward variant in §C. These variants shift call-side behavior, especially for Qwen2.5-VL, but do not reverse the qualitative pattern above: a simple tool-call bonus mainly changes call intensity rather than producing sustained correction on hard tool-free failures. This suggests a design implication of MED: future RL objectives should target selective, successful correction on $\mathcal{D}_{\text{fail}}$ while discouraging Term-3/4-style harms on $\mathcal{D}_{\text{succ}}$, instead of rewarding raw tool frequency. Ultimately, the model learns to **safely coexist** with the tool rather than **master** it. To verify our aggregated curves are

*Table 3.* **Analysis of Schema Interference.** Reported values are average accuracies across all 6 benchmarks (see all results in §E). $Acc_{\text{schema}}$ denotes the setting where the tool schema is provided but *execution is forbidden*. The negative gap ($\Delta = Acc_{\text{schema}} - Acc_{\text{wo}}$) quantifies the intrinsic cost imposed by the tool schema.

| Model | Intrinsic ($Acc_{\text{wo}}$) | Schema-Only ($Acc_{\text{schema}}$) | Gap ($\Delta$) | Tool ($Acc_{\text{w}}$) |
|---|---|---|---|---|
| Qwen2.5-VL | 48.4 | 42.6 | -5.8 | 42.2 |
| Qwen3-VL | 53.0 | 40.0 | -13.0 | 61.2 |

*Table 4.* **Alignment Results.** We employ a strict intersection protocol (Human ∩ Gemini) to identify gains aligned with human logic. Qwen3-VL demonstrates high interpretability, whereas Qwen2.5-VL exhibits a bit of unaligned behaviors.

| Model | Total Samples | Aligned Samples | Aligned Rate |
|---|---|---|---|
| Qwen2.5-VL | 82 | 52 | 63.4% |
| Qwen3-VL | 187 | 174 | 93.0% |

not driven by a single benchmark, we report per-benchmark results for Exp-I–III in §G. We find consistent qualitative behavior across mostly benchmarks and both models, supporting the conclusions drawn from the previous analysis. In addition, we calculate the 95% confidence intervals (CI) for the aggregated results in §H and confirm the stability of the observed results across benchmarks.

## 5. Conclusion and Limitations

In this work, we present a systematic analysis of what vision tool-use RL actually learns in the crop-and-zoom setting. By disentangling intrinsic capability drift, operationalized by the tool-free protocol, from tool-induced effects, and further decomposing tool utility into gain, harm, and their underlying mechanisms, we show that performance improvements in this setting are dominated by intrinsic learning rather than by tool-induced effects. Across models and benchmarks in this setting, vision tool-use RL mainly reduces tool-induced harm, while showing limited improvement in tool contribution. Overall, in the crop-and-zoom setting studied here, vision tool-use RL learns a conservative policy for VLMs that makes tool availability less harmful, but does not reliably extend tool utility beyond the intrinsic hard core.

Limitations include: First, our empirical analysis focuses on a single vision tool, crop-and-zoom; broader validation across genuinely different vision tools and multi-tool settings remains future work. Second, while our analysis suggests MED-inspired reward design as a promising direction, we do not propose a new RL algorithm in this work; designing objectives that explicitly reward selective correction while penalizing tool-induced harm remains future work. Finally, we focus on accuracy; future work could incorporate efficiency, tool-use traces, and more interpretability metrics.

## Impact Statement

This paper presents work whose goal is to advance the field of Large Language Models, specifically in understanding the training dynamics of Vision Tool-Use RL systems. Our analysis contributes to the development of more reliable and interpretable multimodal agents. We do not foresee any immediate negative societal consequences or ethical issues that must be specifically highlighted here.

## Acknowledgements

We acknowledge the Large Language Model Search Algorithms and Reinforcement Learning Research Project of the Generative Artificial Intelligence Research Lab. We also thank Zhengyuan Peng for helpful discussions at the early stage of this project.

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

# A. Derivation of the Four-Term Decomposition

In this section, we provide the formal derivation of the decomposition of the tool-induced gap $G(t)$ presented in Eq. (7).

## A.1. Problem Setup and Partitioning

Recall that $G(t)$ measures the performance gap between the tool-available and tool-free protocols at time $t$:

$$G(t) \triangleq Acc_{\text{w}}(t) - Acc_{\text{wo}}(t). \tag{9}$$

As described in §3.2, we partition the task space $\Omega$ based on the model's intrinsic capability, operationalized as correctness under the tool-free protocol. This yields two disjoint sets at each checkpoint $t$:

- **Failure Set** ($\mathcal{D}_{\text{fail}}$): Samples where the model fails without tools.

- **Success Set** ($\mathcal{D}_{\text{succ}}$): Samples where the model succeeds without tools.

By definition, the tool-free accuracy corresponds exactly to the probability mass of the success set:

$$Acc_{\text{wo}}(t) = P(\mathcal{D}_{\text{succ}}). \tag{10}$$

Consequently, the mass of the failure set is $P(\mathcal{D}_{\text{fail}}) = 1 - Acc_{\text{wo}}(t)$.

## A.2. Expansion of Tool-Available Accuracy

We analyze the tool-available accuracy $Acc_{\text{w}}(t)$ using the Law of Total Probability over the partition $\{\mathcal{D}_{\text{fail}}, \mathcal{D}_{\text{succ}}\}$:

$$Acc_{\text{w}}(t) = P(\mathcal{D}_{\text{fail}})P(\checkmark \mid \mathcal{D}_{\text{fail}}) + P(\mathcal{D}_{\text{succ}})P(\checkmark \mid \mathcal{D}_{\text{succ}}), \tag{11}$$

where $\checkmark$ denotes a correct prediction under the tool-available protocol.

We further expand the conditional success probabilities $P(\checkmark \mid \mathcal{D})$ by conditioning on the tool usage event ($c$: tool called, $\neg c$: tool not called):

$$P(\checkmark \mid \mathcal{D}) = P(c \mid \mathcal{D})P(\checkmark \mid c, \mathcal{D}) + P(\neg c \mid \mathcal{D})P(\checkmark \mid \neg c, \mathcal{D}). \tag{12}$$

Substituting this back into the expression for $Acc_{\text{w}}(t)$:

$$\begin{aligned} Acc_{\text{w}}(t) = &P(\mathcal{D}_{\text{fail}})\left[P(c \mid \mathcal{D}_{\text{fail}})P(\checkmark \mid c, \mathcal{D}_{\text{fail}}) + P(\neg c \mid \mathcal{D}_{\text{fail}})P(\checkmark \mid \neg c, \mathcal{D}_{\text{fail}})\right] \\ &+ P(\mathcal{D}_{\text{succ}})\left[P(c \mid \mathcal{D}_{\text{succ}})P(\checkmark \mid c, \mathcal{D}_{\text{succ}}) + P(\neg c \mid \mathcal{D}_{\text{succ}})P(\checkmark \mid \neg c, \mathcal{D}_{\text{succ}})\right]. \end{aligned} \tag{13}$$

## A.3. Deriving the Tool-Induced Gap $G(t)$

From the definition of $G(t)$ and the property $Acc_{\text{wo}}(t) = P(\mathcal{D}_{\text{succ}})$, we have:

$$G(t) = Acc_{\text{w}}(t) - P(\mathcal{D}_{\text{succ}}). \tag{14}$$

Subtracting $P(\mathcal{D}_{\text{succ}})$ from the expanded form of $Acc_{\text{w}}(t)$ (from Eq. (13)):

$$\begin{aligned} G(t) = &P(\mathcal{D}_{\text{fail}})\left[P(c \mid \mathcal{D}_{\text{fail}})P(\checkmark \mid c, \mathcal{D}_{\text{fail}}) + P(\neg c \mid \mathcal{D}_{\text{fail}})P(\checkmark \mid \neg c, \mathcal{D}_{\text{fail}})\right] \\ &+ P(\mathcal{D}_{\text{succ}})\left[P(c \mid \mathcal{D}_{\text{succ}})P(\checkmark \mid c, \mathcal{D}_{\text{succ}}) + P(\neg c \mid \mathcal{D}_{\text{succ}})P(\checkmark \mid \neg c, \mathcal{D}_{\text{succ}}) - 1\right]. \end{aligned} \tag{15}$$

We focus on the term inside the bracket for the success region $\mathcal{D}_{\text{succ}}$. Using the identity $1 = P(c \mid \mathcal{D}_{\text{succ}}) + P(\neg c \mid \mathcal{D}_{\text{succ}})$, we rewrite the expression:

$$\begin{aligned} &P(c \mid \mathcal{D}_{\text{succ}})P(\checkmark \mid c, \mathcal{D}_{\text{succ}}) + P(\neg c \mid \mathcal{D}_{\text{succ}})P(\checkmark \mid \neg c, \mathcal{D}_{\text{succ}}) - 1 \\ =&P(c \mid \mathcal{D}_{\text{succ}})P(\checkmark \mid c, \mathcal{D}_{\text{succ}}) + P(\neg c \mid \mathcal{D}_{\text{succ}})P(\checkmark \mid \neg c, \mathcal{D}_{\text{succ}}) \\ &- \left(P(c \mid \mathcal{D}_{\text{succ}}) + P(\neg c \mid \mathcal{D}_{\text{succ}})\right) \\ =&P(c \mid \mathcal{D}_{\text{succ}})\left(P(\checkmark \mid c, \mathcal{D}_{\text{succ}}) - 1\right) + P(\neg c \mid \mathcal{D}_{\text{succ}})\left(P(\checkmark \mid \neg c, \mathcal{D}_{\text{succ}}) - 1\right). \end{aligned} \tag{16}$$

Since $P(\checkmark) - 1 = -(1 - P(\checkmark)) = -P(\times)$, this simplifies to:

$$
\begin{aligned}
&= P(c \mid \mathcal{D}_{\text{succ}})\big(- P(\times \mid c, \mathcal{D}_{\text{succ}})\big) + P(\neg c \mid \mathcal{D}_{\text{succ}})\big(- P(\times \mid \neg c, \mathcal{D}_{\text{succ}})\big) \\
&= -P(c \mid \mathcal{D}_{\text{succ}})P(\times \mid c, \mathcal{D}_{\text{succ}}) - P(\neg c \mid \mathcal{D}_{\text{succ}})P(\times \mid \neg c, \mathcal{D}_{\text{succ}}).
\end{aligned}
\tag{17}
$$

Substituting Eq. (17) back into Eq. (15) yields the final four-term decomposition.

### A.4. Final Decomposition

Combining the Gain components (from $\mathcal{D}_{\text{fail}}$) and the derived Harm components (from $\mathcal{D}_{\text{succ}}$), we arrive at the four-term decomposition:

$$
\begin{aligned}
G(t) = &\underbrace{P(\mathcal{D}_{\text{fail}})P(c \mid \mathcal{D}_{\text{fail}})P(\checkmark \mid c, \mathcal{D}_{\text{fail}})}_{\text{Term 1: Call Gain}} \\
&+ \underbrace{P(\mathcal{D}_{\text{fail}})P(\neg c \mid \mathcal{D}_{\text{fail}})P(\checkmark \mid \neg c, \mathcal{D}_{\text{fail}})}_{\text{Term 2: Schema Gain}} \\
&- \underbrace{P(\mathcal{D}_{\text{succ}})P(c \mid \mathcal{D}_{\text{succ}})P(\times \mid c, \mathcal{D}_{\text{succ}})}_{\text{Term 3: Call Harm}} \\
&- \underbrace{P(\mathcal{D}_{\text{succ}})P(\neg c \mid \mathcal{D}_{\text{succ}})P(\times \mid \neg c, \mathcal{D}_{\text{succ}})}_{\text{Term 4: Schema Harm}}.
\end{aligned}
\tag{18}
$$

This derivation confirms that $G(t)$ is the net result of probability mass shifting from intrinsic failures to successes (Gains) minus the mass shifting from intrinsic successes to failures (Harms).

## B. Detailed Experimental Setup

### B.1. Vision Tool.

We focus on the *Crop-and-Zoom tool*, which enables fine-grained visual inspection by extracting high-resolution details from a specified region. The tool is defined by the function *image_crop_and_zoom_in_tool(bbox_2d, label, image_index)*, where `bbox_2d` specifies the region of interest using normalized coordinates in $[0, 1000]$, and `image_index` identifies the target image in multi-image inputs. We select this tool as our primary study target because it is among the most widely adopted vision tools in recent literature (Wang et al., 2025a; Zheng et al., 2025; Su et al., 2025b; Bai et al., 2025a; Zhu et al., 2025). Its prevalence and functional simplicity make it a minimal yet sufficient setting for isolating tool-use learning dynamics without confounding multi-tool interactions.

The full JSON schema used for the Crop-and-Zoom tool is provided below. This schema is injected into the system prompt during both training and inference.

```
tool_schema:
  type: function
  function:
    name: image_crop_and_zoom_in_tool
    description:
      Crops and zooms into a specific region of an image based on
      provided relative coordinates. All coordinate values should
      be float numbers with the range [0, 1000.0].
    parameters:
      type: object
      properties:
        bbox_2d:
          type: array
          items:
            type: number
          minItems: 4
          maxItems: 4
          description:
            Defines the rectangular area for cropping in the format
            [x1, y1, x2, y2]. (x1, y1) marks the upper-left corner,
            and (x2, y2) marks the lower-right corner. All coordinates
            must be in the range [0, 1000] with x1 < x2 and y1 < y2.
        label:
          type: string
          description:
            A descriptive name for the object within the cropped region.
        image_index:
          type: number
          description:
            The index of the image to apply the crop and zoom
            operation on, starting from 0.
      required:
        - bbox_2d
        - label
        - image_index
    strict: True
```

## B.2. Models

We conduct experiments on two widely used VLMs: Qwen2.5-VL-Instruct-7B (Bai et al., 2025b) and Qwen3-VL-Instruct-8B (Bai et al., 2025a). Qwen2.5-VL is widely adopted in recent vision tool-use RL studies, whereas Qwen3-VL represents a stronger, more recent generation of VLMs. While both models support function calling, they differ in prior exposure to the Crop-and-Zoom tool. Qwen2.5-VL has not been explicitly trained with this tool, whereas Qwen3-VL has been explicitly trained with it. This contrast enables a controlled comparison, allowing us to test whether observed training dynamics generalize across different levels of tool familiarity. we fine-tune the LLM backbone while keeping the ViT frozen.

## B.3. Data

We construct a composite dataset of approximately 15k samples samples to train vision tool-use policy for Qwen2.5VL and Qwen3VL. Each sample consists of an image, a question, and a verifiable answer. The core of the dataset is derived from Thyme (Zhang et al., 2025c) and Mini-O3 (Lai et al., 2025), which together account for approximately 80% of the mixture (see Tab. 5 for details). These datasets provide the primary supervision for learning crop-and-zoom tool use in high-resolution settings. To improve robustness and reduce overfitting to specific visual domains, we supplement this core with a diverse collection of samples (~20%) spanning four additional categories: (1) General VQA (e.g., ST-VQA (Biten et al., 2019), LLaVA-OV (Li et al., 2024), EST-VQA (Wang et al., 2020)), (2) Visual Mathematics (MathV-360k (Shi et al., 2024),Video-R1-data (image subset) (Feng et al., 2025), (3) Documents & Charts (RVL-CDIP (Harley et al., 2015), FinChart-Bench (Shu et al., 2025), PlotQA (Methani et al., 2020),SPIQA (Pramanick et al., 2024)), and (4) GUI & Screens (ScreenQA (Baechler et al., 2024)). This diversity promotes a generalized tool-use policy across heterogeneous visual contexts. Tab. 5 details the composition of our RL training dataset. The mixture is dominated by high-resolution VQA tasks

to encourage active tool usage, complemented by a long tail of diverse domains to preserve general capabilities.

### B.4. Decontamination

To prevent data leakage, we apply a visual-similarity-based decontamination procedure. We compute the Perceptual Hash (pHash) for all images in our RL training set and the evaluation benchmarks. Training samples are excluded if those images exhibited a Hamming distance less than 5 with any image in the test sets, following common practice for detecting near-duplicate images. (McKeown & Buchanan, 2023)

### B.5. Training

**Infrastructure.** We conduct all vision tool-use RL training using the verl framework (v0.6.1) (Sheng et al., 2025). The training backend is FSDP2 based on PyTorch v2.8.0 (Paszke et al., 2019) and inference engine is SGLang v0.5.5 (Zheng et al., 2024). The system prompt explicitly includes the tool schema definition, and the maximum conversation turn is 10 (5 user queries and 5 assistant responses).

**Algorithm and Reward Modeling.** We adopt the Group Relative Policy Optimization (GRPO) (Shao et al., 2024) algorithm. Following prior work (Jin et al., 2025), We assign a binary reward $r \in \{0, 1\}$ based solely on final-answer correctness (checking if the content within \boxed{} matches the ground truth). We do not use any additional reward shaping (e.g., tool-call bonuses) and do not apply curriculum learning; all training samples are randomly sampled. Following (Yu et al., 2025b), we use a token-mean loss and omit the KL-divergence penalty to stabilize training.

**Hyperparameters.** The models are trained with a constant learning rate of $1 \times 10^{-6}$ and a global batch size of 256. We set the group size (samples per prompt) to $G = 8$ and the mini-batch size to 32, resulting in an off policy gradient update of 8 (mini batch size $256/32 = 8$). Training proceeds for a total of 1,600 gradient update steps, with checkpoints saved every 80 steps. Tab. 6 provides a complete listing of all hyperparameters.

### B.6. Multi-turn Inference Logic.

Upon generating a tool call, the system parses its arguments. If parsing fails, an error message is returned. If successful, the vision tool executes the crop operation and returns the processed image as the tool response. The tool response is appended to the full conversation history (including all prior images and text) and feed it back to the model. This interaction loop continues until the model outputs a stop token or reaches the maximum turn limit.

### B.7. Evaluation Protocol.

We assess performance on six benchmarks commonly used in vision tool-use RL research (Lai et al., 2025; Zhang et al., 2025c; Hong et al., 2025): VStar (Wu & Xie, 2023), HR-Bench (4K and 8K) (Wang et al., 2025e), and VisualProbe (Easy, Medium, and Hard) (Lai et al., 2025). Together, they cover a range of fine-grained perception and visual understanding tasks. For both tool-available and tool-free inference, we utilize SGLang v0.5.5 (Zheng et al., 2024) with greedy decoding (temperature=0) and a maximum response length of 8,192 tokens. To fully track training dynamics, we evaluate the model at regular intervals (every 80 gradient steps) throughout the 1,600-step training process. This results in 20 checkpoints per model, resulting in a total of 240 evaluation runs (2 models $\times$ 20 ckpts $\times$ 6 tasks).

*Table 5.* **Composition of RL Training Data.** The dataset is primarily sourced from Thyme (Zhang et al., 2025c) and Mini-O3 (Lai et al., 2025) to drive tool learning, supplemented by diverse datasets for domain generalization.

| Data Source | Count | Percentage |
|---|---|---|
| *Primary Vision Tool-Use Sources* | | |
| Thyme-RL | 7,911 | 53.65% |
| Mini-O3 (train) | 4,202 | 28.50% |
| *Diversity Supplements* | | |
| ST-VQA (Biten et al., 2019) | 580 | 3.94% |
| MathV-360k (Shi et al., 2024) | 546 | 3.70% |
| LLaVA-OV (Li et al., 2024) | 382 | 2.59% |
| RVL-CDIP (Harley et al., 2015) | 261 | 1.77% |
| Video-R1-data (Feng et al., 2025) | 201 | 1.36% |
| ScreenQA (Baechler et al., 2024) | 158 | 1.07% |
| FinChart-Bench (Shu et al., 2025) | 136 | 0.92% |
| PlotQA (Methani et al., 2020) | 126 | 0.86% |
| SPIQA (Pramanick et al., 2024) | 124 | 0.84% |
| EST-VQA (Wang et al., 2020) | 118 | 0.80% |
| **Total** | **14,745** | **100.00%** |

*Table 6.* RL Training Hyperparameters and System Configuration.

| Hyperparameter / Setting | Value |
|---|---|
| *Infrastructure* | |
| RL Framework | `verl` v0.6.1 |
| Training Backend | PyTorch v2.8.0 (FSDP2) |
| Inference Engine | SGLang v0.5.5 |
| Compute Precision | `bfloat16` |
| *Algorithm* | |
| Algorithm | GRPO |
| Reward Type | Sparse Outcome Reward (0-1) |
| KL Coefficient ($\beta$) | 0.0 (No explicit KL loss) |
| Loss Objective | Token-mean |
| Clip-high Ratio ($\epsilon - \text{high}$) | 0.28 |
| *Training Recipe* | |
| Learning Rate | $1 \times 10^{-6}$ |
| LR Scheduler | Constant |
| Global Batch Size | 256 |
| Mini-Batch Size | 32 |
| Group Size | 8 |
| Max Response Length | 8192 |
| Max Interaction Turns | 10 (5 User / 5 Assistant) |
| *Optimization Schedule* | |
| Total Training Steps (Gradient Updates) | 1,600 |
| Checkpoint Frequency | Every 80 steps |
| Data Sampling | Random (No Curriculum) |

## C. Tool-Access and Reward Variants

Our main experiments use tool-available GRPO with a binary outcome reward based only on final-answer correctness. To test whether the observed dynamics are specific to this training setup, we compare it with two nearby variants under the same training budget and evaluation protocol: (i) no-tool RL, where tool access is removed during RL training; (ii) the outcome-only tool-RL setting used in the main paper; and (iii) a tool-aware reward variant, which adds a +0.1 bonus to responses that both call the tool and answer correctly.

Fig. 6 shows that these variants change tool-interaction dynamics in a backbone-dependent way. For Qwen2.5-VL, no-tool RL drives both call-side terms close to zero by the final checkpoint, indicating that generic RL alone does not reproduce active tool-interaction learning. Outcome-only tool-RL instead maintains nontrivial Call Gain while reducing harm. Adding the +0.1 tool-aware bonus further increases Call Gain, but also raises Call Harm, suggesting that the bonus primarily changes call-side intensity rather than cleanly improving selective correction. For Qwen3-VL, which starts with stronger prior crop-and-zoom familiarity, the three variants produce much more similar trajectories: Call Gain gradually decreases, Call Harm decreases, and schema-side terms remain small.

Fig. 7 explains where these changes come from. For Qwen2.5-VL, no-tool RL sharply suppresses the call policy on both intrinsic failures and intrinsic successes, so its reduction in Call Harm mainly comes from not calling rather than from better selective tool use. The tool-aware reward has the opposite effect: it substantially increases the call policy on both sides. This raises opportunities for Call Gain, but also exposes more intrinsically solved examples to harmful calls. The conditional correction quality on intrinsic failures improves only modestly and does not produce a qualitatively stronger hard-case correction pattern. For Qwen3-VL, prior crop-and-zoom familiarity makes the call-side trajectories similar across variants, so the simple bonus mainly recalibrates the gain-harm balance rather than inducing a qualitatively different learning pattern.

Overall, the sensitivity to training variants is pronounced for Qwen2.5-VL, whose tool-use behavior is weakly initialized, but much weaker for Qwen3-VL, whose prior crop-and-zoom familiarity makes the call-side trajectories similar across variants. The simple tool-call bonus changes call-side behavior, especially for Qwen2.5-VL, but it does not reverse the main qualitative pattern observed in the outcome-only setting: within this crop-and-zoom setup, training more reliably reduces tool-induced harm than produces sustained gains from correcting tool-free failures. More targeted reward designs may need to reward successful correction on intrinsic failures while penalizing Term-3/4-style harms on intrinsic successes.

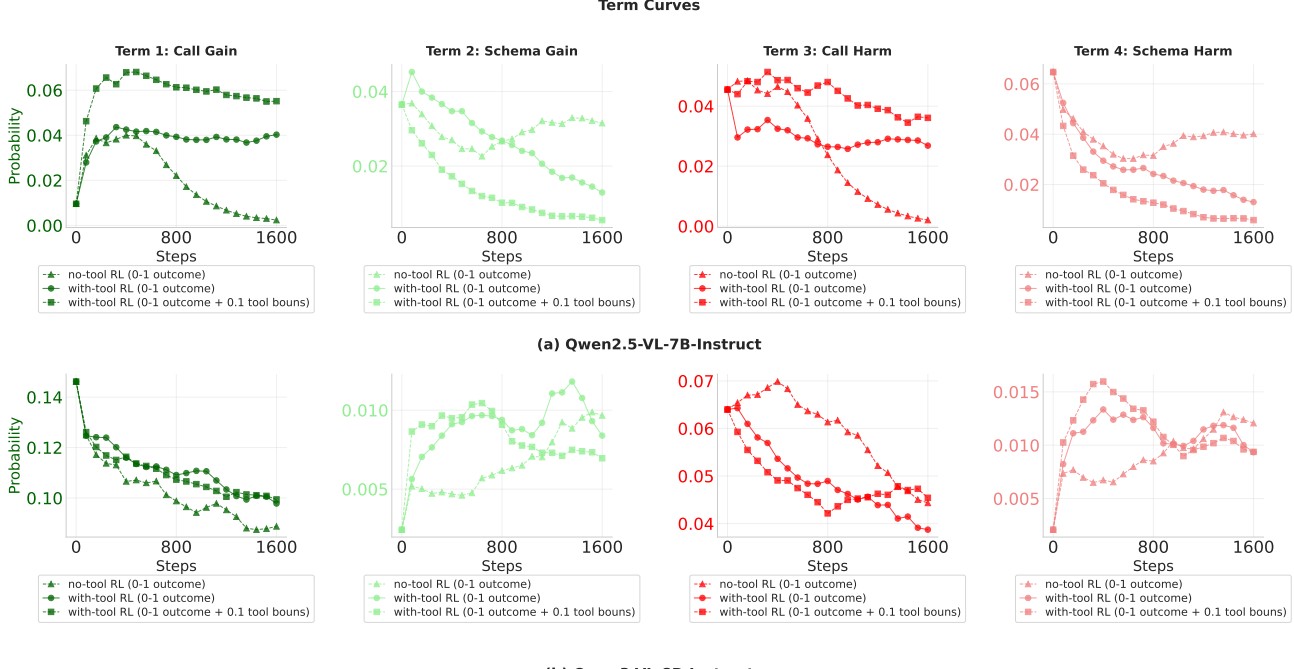

*Figure 6.* **Term-level dynamics under tool-access and reward variants.** We compare no-tool RL, outcome-only tool-RL, and tool-aware reward training on the four MED terms. The tool-aware reward adds a $+0.1$ bonus for correct responses that call the tool.

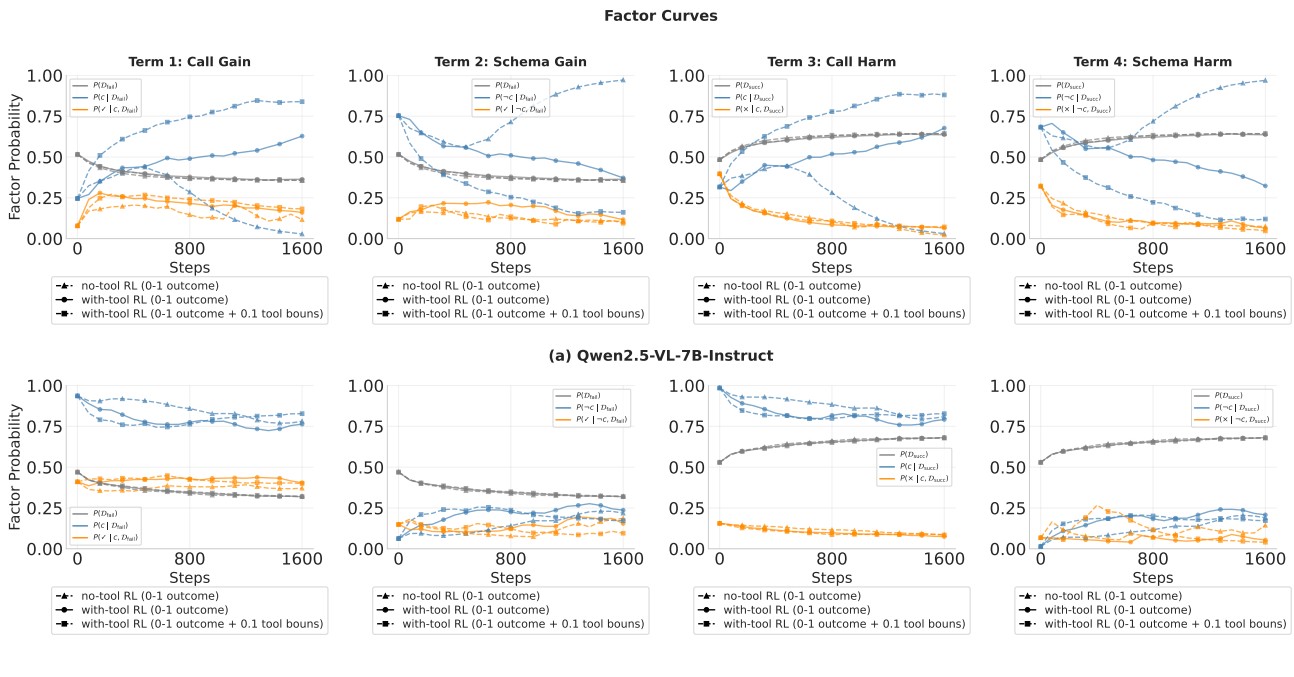

*Figure 7.* **Factor-level dynamics under tool-access and reward variants.** Each MED term is factorized into Mass, Policy, and Quality components. The factor curves show whether term changes arise from the size of the intrinsic success/failure set, the call/no-call policy, or conditional outcome quality.

## D. Data Aggregation and Visualization

We employ two distinct aggregation strategies depending on the visualization objective: *normalized drift aggregation* for relative performance changes, and *direct averaging* for absolute metrics.

### D.1. Normalized Drift Aggregation

This method is used for visualizing relative performance changes (e.g., $\Delta$ Accuracy plots in Fig. 2).

**Step 1: Drift Calculation.** For each benchmark $b$, we compute the performance drift relative to the initial checkpoint:

$$\Delta f_b(t) = f_b(t) - f_b(t_0) \tag{19}$$

where $f_b(t)$ denotes the performance metric at training step $t$ and $t_0$ is the initial step.

**Step 2: Per-benchmark Normalization.** To enable fair cross-benchmark comparison, we normalize each drift by its maximum absolute value:

$$\tilde{f}_b(t) = \frac{\Delta f_b(t)}{\max_t |\Delta f_b(t)|} \tag{20}$$

This maps all drifts to the range $[-1, 1]$ while preserving relative magnitudes within each benchmark. Critically, we apply the same normalization scale to both w/ tool ($f_w$) and w/o tool ($f_{wo}$) scores to preserve the performance gap.

**Step 3: Cross-benchmark Aggregation.** The aggregated performance curve is computed as the arithmetic mean across normalized benchmarks:

$$\bar{f}(t) = \frac{1}{|B|} \sum_{b \in B} \tilde{f}_b(t) \tag{21}$$

where $B$ is the set of benchmarks being aggregated.

### D.2. Direct Averaging

This method is used for absolute metrics that should preserve their physical meaning (e.g., Accuracy plots in Fig. 2, term decompositions in Fig. 3, and factor analyses in Fig. 4).

**Cross-benchmark Aggregation.** For each metric $m$ (e.g., accuracy, term1–4, or probability factors), we directly average the raw values across benchmarks:

$$\bar{m}(t) = \frac{1}{|B|} \sum_{b \in B} m_b(t) \tag{22}$$

No normalization is applied, preserving the absolute scale and interpretability of the metrics.

### D.3. Smoothing

After aggregation (by either method), we apply time-weighted exponential moving average (EMA) to all curves only for visualization clarity:

$$\hat{f}(t_i) = \alpha \cdot \hat{f}(t_{i-1}) + (1 - \alpha) \cdot \bar{f}(t_i) \tag{23}$$

where $\alpha$ is the smoothing factor adjusted by the time interval between consecutive checkpoints.

### D.4. Area-based Metrics

For quantifying cumulative magnitude of performance changes, we use trapezoidal integration over the raw curves.

**Tool Impact Decomposition.** We separately integrate positive and negative contributions:

$$|B_{\Delta_{\text{tool}}}|^+ = \int_{t_0}^{T} \max(0, \hat{f}_w(t) - \hat{f}_{wo}(t)) \, dt \tag{24}$$

$$|B_{\Delta_{\text{tool}}}|^- = \int_{t_0}^{T} \min(0, \hat{f}_w(t) - \hat{f}_{wo}(t)) \, dt \tag{25}$$

where $\hat{f}_w$ and $\hat{f}_{wo}$ are the raw w/ and w/o tool $\Delta$ accuracy curves.

**Tool Contribution Ratio.** We define $S_{\text{tool}}$ as the fraction of total performance change attributable to tool usage:

$$S_{\text{tool}} = \frac{|B_{\Delta_{\text{tool}}}|^+ + |B_{\Delta_{\text{tool}}}|^-}{|B_{\text{wo}}| + |B_{\Delta_{\text{tool}}}|^+ + |B_{\Delta_{\text{tool}}}|^-} \tag{26}$$

where $B_{\text{wo}} = \int_{t_0}^{T} |\hat{f}_{wo}(t)| \, dt$ quantifies the intrinsic performance change without tools.

# E. Detailed Analysis of Schema Interference

*Table 7.* **Per-Task Result of Schema Interference.** We report the performance of two models across all 6 tasks. "Interference Gap" quantifies the performance degradation caused by providing the tool schema while forbidding execution ($\Delta = Acc_{\text{schema}} - Acc_{\text{wo}}$). The consistently negative gaps highlight the intrinsic cost imposed by the schema.

| Setting | VStar | HRBench (4k) | HRBench (8k) | VisualProbe (Easy) | VisualProbe (Medium) | VisualProbe (Hard) | Avg. |
|---|---|---|---|---|---|---|---|
| ***Qwen2.5-VL-Instruct-7B*** | | | | | | | |
| (A) Intrinsic ($Acc_{\text{wo}}$) | 78.0 | 69.2 | 64.9 | 39.0 | 16.4 | 22.6 | 48.4 |
| (B) Distracted ($Acc_{\text{schema}}$) | 74.3 | 66.9 | 61.6 | 24.8 | 14.6 | 13.2 | 42.6 |
| (C) Tool-Available ($Acc_{\text{w}}$) | 74.9 | 70.6 | 62.5 | 21.3 | 12.7 | 11.3 | 42.2 |
| **Interference Gap (B - A)** | -3.7 | -2.3 | -3.3 | -14.2 | -1.8 | -9.4 | -5.8 |
| ***Qwen3-VL-Instruct-8B*** | | | | | | | |
| (A) Intrinsic ($Acc_{\text{wo}}$) | 82.7 | 74.4 | 71.0 | 41.8 | 23.5 | 24.5 | 53.0 |
| (B) Distracted ($Acc_{\text{schema}}$) | 57.1 | 64.4 | 56.8 | 27.0 | 16.8 | 17.9 | 40.0 |
| (C) Tool-Available ($Acc_{\text{w}}$) | 90.1 | 79.5 | 72.4 | 56.7 | 37.7 | 31.1 | 61.2 |
| **Interference Gap (B - A)** | -25.6 | -10.0 | -14.2 | -14.8 | -6.7 | -6.6 | -13.0 |

## F. Evaluating Alignment of Tool Call Gains with Human Reasoning

**Analysis Prompt for Gemini3-Pro-Preview**

```
You are an expert in visual question answering. I will show you:
1. The original image
2. One or More cropped regions from that image
3. The question that needs to be answered
4. The ground truth answer
5. The model's response process

Your task is to evaluate whether the cropped region in the tool call is beneficial

for the model to answer the question during its response process.

Question: {question}

Ground Truth Answer: {answer}

Model's Response Process:
{response}

Please analyze:

1. Does the cropped region contain relevant information for answering the question?
2. Did the crop help the model in its reasoning process to arrive at the answer?
3. Was the crop well-positioned and properly sized for this task?

Respond with a JSON object with the following format:
{{
    "is_beneficial": true/false,
    "relevance_score": <1-5, where 5 is highly relevant>,
    "crop_quality_score": <1-5, where 5 is perfectly cropped>,
    "reasoning": "<your detailed explanation>"
}}

Only output the JSON, no other text.
```

We built a Streamlit labeling interface. Human annotators follow the same prompt as above but only answer: "Is the tool call beneficial?". The final labels in Tab. 4 are the intersection of human and Gemini results.

## G. Per-Benchmark Results for Exp-I–III

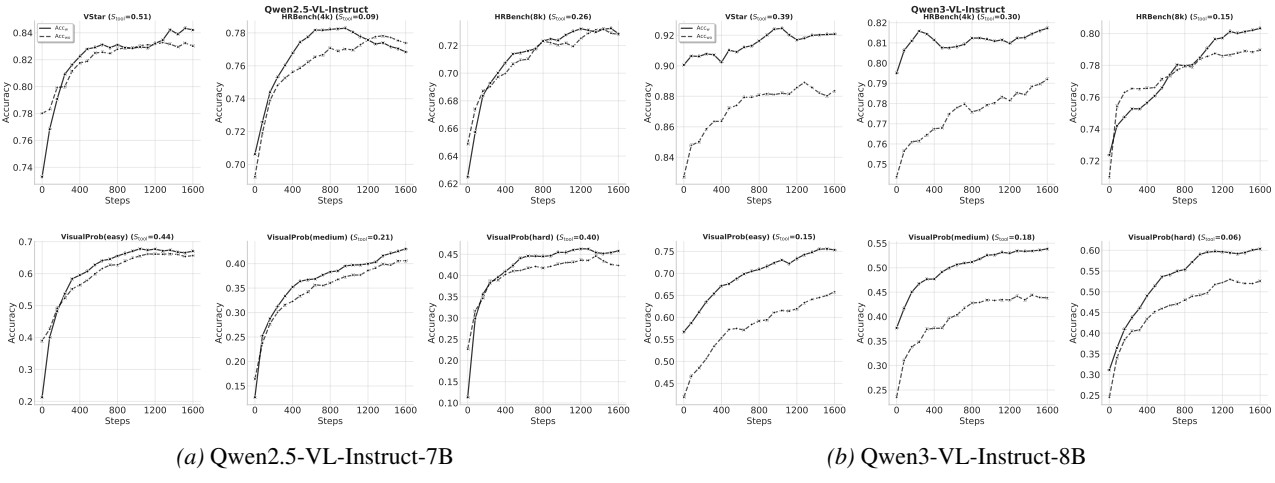

*(a)* Qwen2.5-VL-Instruct-7B                *(b)* Qwen3-VL-Instruct-8B

*Figure 8.* Quantifying Intrinsic and Tool-Induced Drift

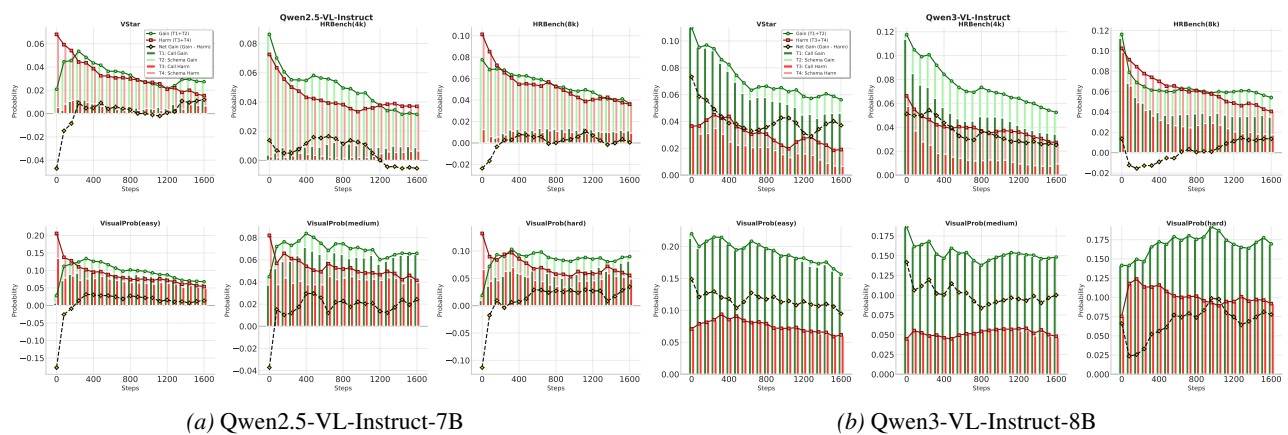

*(a)* Qwen2.5-VL-Instruct-7B        *(b)* Qwen3-VL-Instruct-8B

*Figure 9.* Decomposition of Tool-Induced Performance Gap

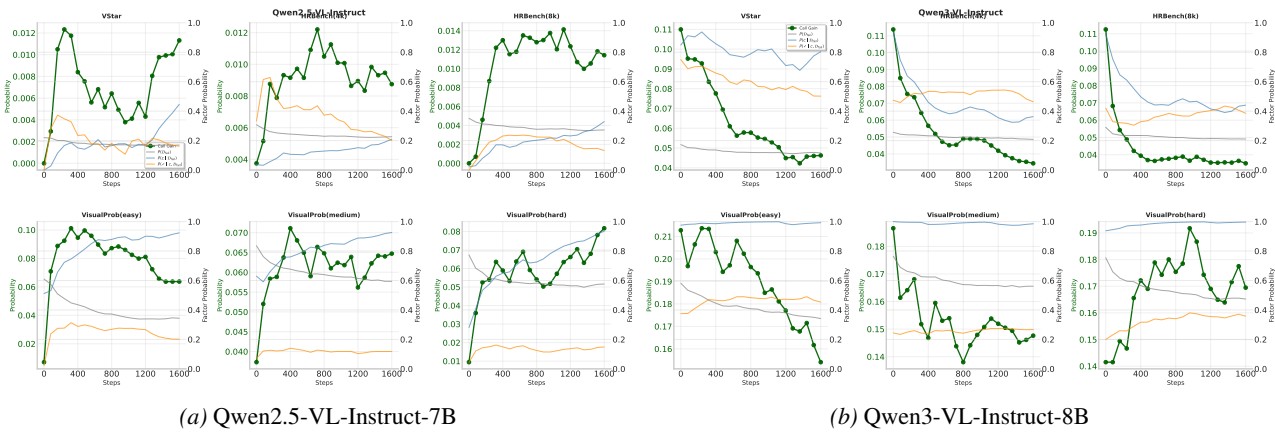

*(a)* Qwen2.5-VL-Instruct-7B        *(b)* Qwen3-VL-Instruct-8B

*Figure 10.* Factor Decomposition of Tool-Induced Effects (Term1)

## H. Confidence Intervals for Aggregated Results

To verify the reliability of the observed trends in the aggregated results, we compute the *95% confidence intervals (CI)* for key performance metrics. The confidence intervals are calculated using a paired bootstrap method. For each checkpoint, we resample 1000 times, computing the accuracy and other relevant metrics for each resample, and then report the CI based on these resampled values. The table below presents the aggregated results for two models: Qwen2.5-VL and Qwen3-VL. Each model is evaluated across multiple checkpoints, with both initial and final performance values, along with their 95% CI:

*Table 8.* Key Evaluation Metrics (with 95% Confidence Intervals)

| Model | Acc (w/o tool) | | Acc (w/ tool) | | $S_{tool}$ | $P(\checkmark \mid c, \mathcal{D}_{fail})$ | | $P(\times \mid c, \mathcal{D}_{succ})$ | |
|---|---|---|---|---|---|---|---|---|---|
| | Init (before train) | Final (after train) | Init (before train) | Final (after train) | | Init (before train) | Final (after train) | Init (before train) | Final (after train) |
| Qwen2.5-VL | 0.4836 [0.4607, 0.5070] | 0.6336 [0.6063, 0.6587] | 0.4193 [0.3977, 0.4405] | 0.6535 [0.6280, 0.6795] | 0.3074 [0.2037, 0.4134] | 0.0967 [0.0257, 0.1683] | 0.1324 [0.0954, 0.1724] | 0.3982 [0.2979, 0.5170] | 0.0438 [0.0260, 0.0630] |
| Qwen3VL | 0.5299 [0.5061, 0.5530] | 0.6931 [0.6679, 0.7167] | 0.6124 [0.5899, 0.6361] | 0.7423 [0.7187, 0.7657] | 0.2108 [0.1042, 0.3094] | 0.4105 [0.3700, 0.4480] | 0.3658 [0.3112, 0.4255] | 0.1567 [0.1194, 0.1971] | 0.0680 [0.0483, 0.0902] |

