# OpenReview forum: "What Does Vision Tool-Use Reinforcement Learning Really Learn? Disentangling Tool-Induced and Intrinsic Effects for Crop-and-Zoom"
_ICML.cc/2026/Conference — ICML 2026 regular_

### Official Review · Reviewer_Gc5V · 2026-03-01

**Soundness:** 3
**Presentation:** 3
**Significance:** 3
**Originality:** 3
**Overall Recommendation:** 4
**Confidence:** 3

**Summary:**

The paper investigates what tool augmented vision language models learn during training. To achieve this, the paper develops an analytical framework that breaks down the model improvements into different components (e.g., improvements resulting from correct tool calls). The paper analyzes two VLMs on multiple datasets and suggests that the performance gains are mainly due to improvements in the intrinsic capabilities of the model and improvements in tool calling capabilities have a smaller role in the overall boost in performance. Also, the presented analysis suggests that during training, RL mostly learns not to degrade the model performance in the presence of tools rather than learning to use them effectively (i.e., co-exist with tools).

**Compliance With Llm Reviewing Policy:**

Affirmed.

**Final Justification:**

The rebuttal has answered my main question about the experimental setup, which impacted the soundness of the paper.
Moreover, the authors mentioned that they will move a lot of the derivation of their main equation to the appendix, which should significantly improve readability.

**Key Questions For Authors:**

See the weaknesses and strengths section.

**Limitations:**

yes

**Strengths And Weaknesses:**

# Strengths

1. The paper is well structured and the goals and contributions of the paper are clearly presented.
2. Separating tool use capabilities from models’ intrinsic capabilities is very interesting. Moreover, it is an important problem to understand what models learn during training and the proposed analysis approach could be useful for future work.
3. The authors conduct extensive experiments on two models and several datasets which help better understand the value of the proposed approach.





# Weaknesses


1. The paper compares the boost in performance during training with and without tools. Where "f_w" is the boost in performance at step t (accuracy(t) - accuracy(0)) when using tools and "f_wo" is the boost in performance when not using tools during training. When f_w > f_wo, the paper assumes that for the tool equipped model, the difference in performance boost (i.e., “f_w - f_wo”) is because of the improvement to tool-related skills of the model and the rest (i.e., “f_wo” portion of the performance boost) is because of the improvements in the intrinsic/general capabilities of the model. However, there is no supporting argument to justify this assumption. While the model without tools gets a boost in performance of "f_wo", there is no guarantee that the model with tools would learn the same thing (and thus no guarantee that "f_wo" portion of the boost it achieves are because of intrinsic improvements). For example, one possibility is that the model with tools might be worse in learning intrinsic features and a smaller portion of its boost in performance is due to better intrinsic capabilities. Since the contributions and arguments in the paper heavily rely on this assumption, it is important to at least discuss why the contribution of intrinsic capabilities to the improvements of the model with access to tools is assumed to be similar to the contribution of better intrinsic capabilities to the boost in performance for the model without access to tools.

2. The “schema gain” part of the analysis represents samples that the initial model fails to solve. But A) after training with tools, the model succeeds on these samples; and B) the model does not call a tool for these samples even after training. The paper claims that improvements in these scenarios are just due to the presence of the tool schema in the prompt. However, there is no supporting discussion for this claim. For example, it is possible that the model with access to crop-and-zoom tool uses this tool during the training to look more closely at regions of images and as a result, its intrinsic skills also improve because of the more detailed inspection of samples. Therefore, it is possible that what the paper calls “Schema Gain” is actually a gain because of improvements to the intrinsic capabilities of the model as a result of training with tools.


3. Deriving equation 7 is a very big part of the paper and not easy to follow. The equation shows that the performance gains from training in the presence of tools can be divided into the following four parts:

    - (positive contribution) Boost in performance resulting from initially failed examples that are successful after training by **using** the tool at inference time.
    - (positive contribution) Boost in performance resulting from initially failed examples that are successful after training **without** using the tool at inference time.
    - (negative contribution) degradation in performance resulting from initially successful examples that are failing after training **using** the tool at inference time
    - (negative contribution) degradation in performance resulting from initially successful examples that are failing after training **without** using the tool at inference time

This is a very intuitive breakdown of the gains into the four possible components. The breakdown itself is useful. But, how it is currently presented through mathematical derivation makes the paper difficult to understand and I am not convinced of the added value it provides.


**Optional Suggestion**: it would be helpful if the authors could discuss in the paper how the presented analysis and findings could benefit future work, ideally with examples of future work directions or applications.

---

> ### Author Rebuttal · Authors · 2026-03-31
>
> ## W1: The decomposition may implicitly assume that the tool-free performance improvement over training can be interpreted as intrinsic/general improvement, even though tool exposure during RL may also affect this quantity.
>
> A: To clarify, the core MED decomposition in the paper is based on paired evaluation of the same tool-RL checkpoint: `Acc_w(t)` and `Acc_wo(t)` are the tool-available and tool-free accuracies of the same model checkpoint, not two separately trained models. In the paper, `Acc_wo` denotes the model's performance under tool-free inference at the same vision tool-use RL checkpoint. We agree that this is a definition within our setting, not a perfectly isolated, tool-independent capability measure, and we therefore use it as a practical tool-free reference rather than as a pure intrinsic measurement.
>
> Two pieces of evidence support this choice.
>
> First, the more obvious alternative, showing the schema while forbidding execution, is itself a more intrusive and less natural probe: the model sees the tool schema but is artificially prevented from using it. In our schema-interference check (Tab. 1 & 5), this protocol reduces accuracy by 5.8 points on Qwen2.5-VL and 13.0 points on Qwen3-VL, indicating that this is not a better reference setting.
>
> Second, for rebuttal we additionally ran a no-tool RL baseline `[url, Fig.II]`: the final tool-free performance remains very close to outcome-only tool-RL on both backbones (Qwen2.5-VL: 0.644 vs. 0.636; Qwen3-VL: 0.678 vs. 0.681), and the mean absolute per-checkpoint `w/o tool` gap over all 21 checkpoints is only 0.0175 and 0.0191, respectively. These results support using `Acc_wo` as a practical tool-free reference. We will therefore revise the wording accordingly: in the paper, `Acc_wo` is our practical tool-free reference rather than a perfectly isolated intrinsic measure.
>
> ## W2: Term 2 ("Schema Gain") may over-attribute no-call recovery to the schema itself, even though such recovery could also reflect abilities acquired during tool-based RL training.
>
> A: We agree that the label may overstate what is identified. In the paper, Term 2 is formally defined in Eq. 7 as `P(D_fail(t)) P(¬c | D_fail(t)) P(✓ | ¬c, D_fail(t))`: after defining `D_fail(t)` under the tool-free protocol, it captures cases that become correct under the tool-available protocol without actually calling the tool. A more precise description is therefore schema-conditioned no-call gain.
>
> We do not intend this as a claim that the gain is purely caused by the schema itself. Because Term 2 is defined on `D_fail(t)`, these cases are still failures under the tool-free protocol at the same checkpoint; the caution is about how to interpret their recovery under the tool-available no-call condition. As you note, tool-based training could also improve these no-call cases through other mechanisms. We will revise the wording to make this boundary explicit. Even with this more cautious interpretation, the decomposition remains useful because it still separates recovery with calls (Term 1) from no-call recovery under the tool-available protocol (Term 2).
>
> ## W3 & Optional Suggestion: Eq.7's four-way decomposition is intuitive, but the current derivation is hard to follow and its added value is not sufficiently clear.
>
> A: We agree the presentation can be more step-by-step. We will make the derivation clearer in the main text and move more of the algebraic detail to the appendix.
>
> The value of Eq.7 is not the algebra itself, but that it turns the overall tool-available vs. tool-free gap into four interpretable cells along two axes: gain vs. harm, and call vs. no-call. This is what enables Exp-II, Exp-III, and the later Mass/Policy/Quality diagnosis. More importantly, it makes future work more targeted: instead of only asking whether tool-RL improves final accuracy, one can ask whether a new method reduces harm, increases call-based recovery, or changes no-call behavior, which is useful for designing better rewards, better calling policies, and more targeted ablations/interventions.
>
> This also speaks to your helpful suggestion about how the analysis can guide future work. For example, if a future reward design improves final accuracy, Eq.7 can reveal whether the gain comes mainly from reducing call-induced harm (Term 3) or from increasing recovery through tool calls (Term 1); with the later diagnosis, one can further tell whether higher Term 1 comes from a higher call rate on `D_fail` or better correction quality `P(✓ | c, D_fail)`. One concrete direction is a GRPO-style variant that performs paired tool-free/tool-available rollouts within each sample group and uses MED-inspired reward shaping to encourage correction on `D_fail` while discouraging Term-3/4 harms on `D_succ`.
>
> url: https://anonymous.4open.science/r/anonymous-6124

---

> > ### Author Rebuttal · Reviewer_Gc5V · 2026-04-04
> >
> > Thank you for the detailed response. I have updated my score.

---

> > > ### Author Response · Authors · 2026-04-04
> > >
> > > Thank you for the thoughtful review and follow-up. We appreciate that our rebuttal clarifications addressed your concerns, and we are grateful for your updated assessment. We will incorporate these clarifications into the paper, including the more careful wording around `Acc_wo`, the interpretation of Term 2, and the presentation of Eq.7.

---

### Official Review · Reviewer_viaB · 2026-03-07

**Soundness:** 3
**Presentation:** 3
**Significance:** 3
**Originality:** 3
**Overall Recommendation:** 4
**Confidence:** 4

**Summary:**

This paper investigates the training dynamics of vision tool-use RL. The authors introduce MED, a decomposition framework that separates intrinsic drift from tool-induced drift and further analyzes different gain/harm components of tool use. Experiments on crop-and-zoom RL with two Qwen-VL backbones suggest that the main effect of training is not so much to improve tool-derived correction, but rather to improve the model’s intrinsic performance and reduce harm from tool interaction.

**Compliance With Llm Reviewing Policy:**

Affirmed.

**Final Justification:**

My concerns have been adequately addressed.

**Key Questions For Authors:**

1. Can the authors include a no-tool RL baseline? This would be important for separating general RL gains from gains that specifically come from tool interaction.

2. Can the authors provide a more detailed error analysis of the remaining failures? For example, what kinds of medium-difficulty or persistent-failure cases remain unsolved, and are these cases associated with factors such as small answer regions, localization errors, or inability to internalize tool observations?

3. Have the authors considered modifying the reward design to explicitly encourage useful tool calls? With the current reward setup, the model may simply become more conservative and avoid using the tool later in training.

**Limitations:**

Yes

**Strengths And Weaknesses:**

Strengths

1. The paper asks a genuinely interesting question: what vision tool-use RL is actually learning, rather than simply reporting end-task gains. I think this is a useful and timely direction.

2. The proposed MED framework is fairly novel and the analysis is more insightful than a standard benchmark paper. In particular, separating intrinsic drift from tool-induced effects, and then decomposing gain/harm, is a thoughtful way to study training dynamics.

3. The empirical study is reasonably careful within its chosen setup: the paper tracks checkpoints over training, evaluates on six benchmarks, and compares two backbones with different initial tool priors. This gives the analysis more depth than a single final-score comparison.

Weaknesses

1. The paper’s main conclusion may be somewhat broader than what the current experiments fully support. The study is centered on a crop-and-zoom tool, so it is not yet clear how well the same conclusion would transfer to other kinds of vision tools.

2. A no-tool RL baseline would strengthen the paper. The current comparison is already informative, but it still does not fully separate gains from general RL post-training and gains that may come from learning through tool interaction.

3. The paper would benefit from more detailed error analysis. In particular, it would be useful to know what kinds of cases remain difficult after training, and whether these failures are related to factors such as localization difficulty or answer-region size.

4. It would also be helpful to test whether the current conclusion is partly shaped by the reward design. Since the training seems to rely mainly on final-answer correctness, the observed tendency toward conservative tool use may be specific to this setup.

---

> ### Author Rebuttal · Authors · 2026-03-31
>
> ## W1: The main conclusion may be broader than what the current crop-and-zoom experiments can fully support; how far should these findings generalize to other vision tools?
>
> A: We appreciate this suggestion, and we will clarify the intended scope more explicitly: empirically, our findings are best interpreted within the crop-and-zoom setting we study, rather than as an established result for all vision tools. We chose crop-and-zoom because it is one of the most widely used visual tools in recent vision tool-use RL [1-3], making it a well-motivated focal setting. MED itself is more general: it only requires paired tool-free/tool-available evaluation on the same checkpoint plus observable call/no-call behavior. We will revise the wording to make this distinction explicit: the empirical findings are for crop-and-zoom, while MED is the more general contribution.
>
> ## W2&Q1: Can the authors include a no-tool RL baseline to better separate generic RL gains from gains that arise through tool interaction?
>
> A: Yes. We added a no-tool RL baseline on both backbones, with the same 200 steps and evaluation pipeline but no tool access during RL training `[url, Fig.II]`. Under the same tool-free evaluation, no-tool RL and with-tool RL produce broadly similar gains in `Acc_wo`. However, with-tool RL yields clearly larger `Acc_w` and a consistently larger `G(t)=Acc_w-Acc_wo`; on Qwen2.5-VL it turns `G(t)` positive while no-tool RL remains negative. Diagnose explains why `[url, Fig.I(b)]`: on Qwen2.5-VL, no-tool RL drives near-zero calling (`P(c|D_fail)`: 0.2466 -> 0.0283; `P(c|D_succ)`: 0.3167 -> 0.0306), collapsing call-side gain and harm. The same direction holds on Qwen3-VL with a smaller gap due to its stronger prior tool familiarity. Thus, similar `Acc_wo` gains can arise without tool exposure, but the larger `Acc_w` and `G(t)` under with-tool RL still show additional gains from tool interaction.
>
> ## W3&Q2: Can the authors provide a more detailed error analysis of the remaining failures, especially whether medium-difficulty or persistent failures are tied to localization difficulty, small answer regions, or failure to internalize tool observations?
>
> A: Yes. We added a manual case analysis by two authors on Qwen2.5-VL at the final checkpoint. We examine persistent tool-free failures `D_fail(0)\cap D_fail(t_final)` on VStar, HR-Bench 4K, and VisualProbe Hard. Within this cohort, 370 cases still fail under tool-available evaluation: 271 no-call failures and 99 call-but-still-wrong failures. Two authors label all 99 called failures and a random sample of 100 no-call failures. This complements Fig.5 in paper, where `P(✓|c,D_fail)` improves on the fixed initial cohort but remains limited on current and persistent failures.
>
> | Failure group | Breakdown |
> | --- | --- |
> | no-call + wrong (`n=100`) | should have called: 82; not clearly a missed call: 18 |
> | call + wrong (`n=99`) | crop wrong: 52; crop right but answer wrong: 37; crop right but still hard: 10 |
>
> In this sampled taxonomy, the dominant modes are missed calls, wrong crops, weak use of tool observations, and cases where the crop is correct but the answer region remains too small or visually hard. Representative cases are in `[url, Fig.III]`.
>
> ## W4&Q3: Have the authors considered modifying the reward design to explicitly encourage useful tool calls? With the current reward setup, the model may simply become more conservative and avoid using the tool later in training.
>
> A: Yes. We tested a tool-aware reward variant on both backbones, with the same 200-step pipeline, adding a +0.1 bonus to responses that both call the tool and answer correctly `[url, Fig.I]`. We also note that recent vision tool-use RL work often relies on outcome-based rewards rather than explicit tool-aware shaping [2]. We do not think the outcome-only baseline simply teaches tool avoidance (Fig.4 in paper): on Qwen2.5-VL, `P(c|D_fail)` and Term 1 rise over training; on Qwen3-VL, training mainly recalibrates an initially near-1 call rate. The bonus does change the dynamics, but mainly through call frequency rather than better selective tool use. On Qwen2.5-VL, it pushes `P(c|D_fail)`/`P(c|D_succ)` from 0.6280/0.6771 to 0.8387/0.8801, and final `Gain - Harm` changes only modestly (0.0134 under outcome-only reward vs. 0.0186 under tool-aware reward). On Qwen3-VL, gain is 0.1062 vs. 0.1064 while harm rises from 0.0480 to 0.0548, worsening final `Gain - Harm` from 0.0582 to 0.0516. Importantly, the bonus does not reverse the harm-side trend from Exp-III: `P(×|c,D_succ)` still declines substantially over training. Thus, reward design matters, but it does not overturn the qualitative picture in Exp-III: in the current crop-and-zoom setting we study, tool-use RL suppresses conditional harm more reliably than it enhances correction on hard failures.
>
> url: https://anonymous.4open.science/r/anonymous-6124
>
> [1] Pixel Reasoner. NeurIPS 2025
>
> [2] DeepEyesV2. ICLR 2026
>
> [3] Mini-o3. ICLR 2026

---

> > ### Author Rebuttal · Reviewer_viaB · 2026-04-03
> >
> > The authors have addressed my main concerns with additional experiments and analysis. A minor clarification that could further improve the paper is to more clearly explain the distinction between “crop right but still hard” and “crop right but answer wrong” in the error analysis.

---

> > > ### Author Response · Authors · 2026-04-03
> > >
> > > Thank you for the helpful follow-up. We agree this distinction should be stated more explicitly in the error analysis.
> > >
> > > In our manual taxonomy under human inspection, the difference is:
> > >
> > > | Category | Operational meaning | Main failure mode | Representative case |
> > > | --- | --- | --- | --- |
> > > | crop right but answer wrong | The crop correctly localizes the relevant region, and under human inspection the cropped view is sufficient to determine the answer. | The cropped view appears sufficient under human inspection, yet the model still answers incorrectly. | [url, Fig.III, Case 3] |
> > > | crop right but still hard | The crop correctly localizes the relevant region, but under human inspection the cropped view is still insufficient for confident answering because the evidence remains too small, ambiguous, or visually difficult even after zooming. | The cropped evidence is still too hard to read even after correct localization. | [url, Fig.III, Case 4] |
> > >
> > > Concretely, in Case 3, the crop already makes the butterfly body visibly brown under human inspection, so the remaining error is not localization. In Case 4, even after zooming, it remains difficult to tell which arrow on the sign corresponds to the Zaragoza text, so the crop is correct but the evidence is still hard to read.
> > >
> > > So the key distinction is that both categories involve cases where the model localizes the right region but still answers incorrectly, and we further separate them by how answerable the cropped region is under human inspection: in "crop right but answer wrong," a human can answer from the crop, whereas in "crop right but still hard," the model answers incorrectly and the crop remains hard even for a human to read confidently. This is a descriptive manual distinction under human inspection, and we will make it explicit in the error-analysis description.
> > >
> > > url: https://anonymous.4open.science/r/anonymous-6124

---

### Official Review · Reviewer_NwJo · 2026-03-10

**Soundness:** 3
**Presentation:** 4
**Significance:** 2
**Originality:** 3
**Overall Recommendation:** 5
**Confidence:** 2

**Summary:**

This paper investigates what vision tool-use RL actually learns when equipping VLMs with a crop-and-zoom tool. While prior work reports performance gains from tool-use RL, this paper claims that it remains unclear whether these gains arise from improved tool mastery or from simply better intrinsic reasoning. The authors propose MED (Measure-Explain-Diagnose), an analytical framework that disentangles intrinsic performance drift (tool-free accuracy changes) from tool-induced effects (the gap between tool-available and tool-free performance). They further decompose the tool-induced gap into four probabilistic components (Call Gain, Schema Gain, Call Harm, and Schema Harm) and factor each into Mass, Policy, and Quality terms to diagnose training dynamics. Across 2 VLM backbones (one tool-naive and one tool-native) and 6 high-resolution benchmarks, they find that most gains stem from intrinsic learning rather than improved tool execution. The paper concludes that current vision tool-use RL learns to safely coexist with tools rather than master them.

**Compliance With Llm Reviewing Policy:**

Affirmed.

**Final Justification:**

The rebuttal addressed my concerns. I am probably at a 4.5, but wanted to give the boost (while noting that my confidence is still low) because I do think the paper is a good contribution, albeit a bit limited to one tool use domain.

**Key Questions For Authors:**

- How can these findings generalize beyond this one tool that was evaluated?
- Can you please rebut my above point about intrinsic performance definition? Why is it acceptable for the model to see the tool schema at RL training time?
- Have you tried forced interventions, like call the tool every x timesteps, or if no progress for y timesteps call the tool, etc?

**Limitations:**

yes

**Strengths And Weaknesses:**

Strengths:
- The paper addresses important question in tool-use RL and whether its benefits are genuinely due to tool mastery or merely strengthens intrinsic capability.
- The 4 terms in Eq.7 are well motivated and the decomposition seems sound.
- Rather than reporting only final metrics, the paper analyzes trajectories over training. This dynamic perspective strengthens the attribution claims and provides more mechanistic insight.
- The use of 2 backbones with different levels of prior tool familiarity (tool-naive vs tool-native) is well-motivated and strengthens the generality of the observations. The analysis includes robustness checks (e.g. moving failure sets, confidence intervals).

Weaknesses:
- The paper has limited tool and training regime scope. The study focuses on a single tool and a single RL setup with sparse binary rewards. It remains unclear whether the observed dynamics generalize to multi-tool environments or to settings with explicit tool-aware reward shaping. Stronger claims about "vision tool-use RL" broadly would require broader coverage. This weaknesses is explicitly pointed out in the limitations, and I argue that it is a rather major one.
- Intrinsic performance is defined as tool-free inference, but after already training with the tool schema present during RL. Because the model is always exposed to the schema during RL training, tool-free evaluation may not perfectly isolate intrinsic capability. While the authors justify their choice, the isolation is not fully clean to me.
- I think the paper's diagnostic would be stronger if the authors performed causal interventions like reward shaping ablations or forced call evaluations, to more definitively isolate tool-learning effects.
- (Minor) although the benchmarks are high-resolution and commonly used, it is not fully established whether they fundamentally require tool use for solving the hardest cases. If intrinsic scaling alone can solve most examples, intrinsic dominance may be expected.

---

> ### Author Rebuttal · Authors · 2026-03-31
>
> ## W1&W4&Q1: How can these findings generalize beyond this one tool that was evaluated?
>
> A: We appreciate this concern, and will clarify the intended scope more explicitly: empirically, our findings are best interpreted within the single-tool crop-and-zoom setting we study, rather than as an established claim about arbitrary vision tools or multi-tool environments. Crop-and-zoom is a widely used and practically important visual operator in recent vision tool-use RL [1-4], and our benchmark suite is standard in recent crop-and-zoom / visual-search work.
>
> MED itself is more general, since it only requires paired tool-free/tool-available evaluation plus observable call/no-call behavior; this paper instantiates it in a representative crop-and-zoom setting with two backbones, six benchmarks, 21 checkpoints, and robustness checks. On the training setup, we also tested a tool-aware reward variant `[url, Fig.I 0-1 outcome + 0.1 tool bonus]`. It changes call frequency and shifts the gain-harm tradeoff, but it does not change our main empirical finding: even with explicit tool-call rewards, training still reduces tool-induced harm more reliably than it turns tool calls into strong correction on hard cases. We will revise the wording to keep this empirical scope explicit and avoid broader claims beyond it.
>
> ## W2&Q2: Why is it acceptable to use tool-free inference as the intrinsic reference if the model still sees the tool schema during RL training?
>
> A: We agree that this is not a perfectly clean isolation, and we will revise the wording to present `Acc_wo` as a practical tool-free reference rather than a pure intrinsic quantity. We did not use the schema-shown-but-execution-forbidden setting as the reference because it is a more intrusive and less natural intervention: the model sees the tool schema but is artificially prevented from using it. In our schema-interference check (Tab. 1 & 5), this protocol reduces accuracy by 5.8 points on Qwen2.5-VL and 13.0 points on Qwen3-VL, indicating that this is not a better reference setting.
>
> Our additional no-tool RL baseline `[url, Fig.II no-tool RL]` further supports using `Acc_wo` as a practical tool-free reference: the tool-free trajectory under with-tool outcome-only RL remains close to that of no-tool RL, with nearly identical final values on both backbones (Qwen2.5-VL: 0.636 vs. 0.644; Qwen3-VL: 0.681 vs. 0.678). The table below reports the mean absolute per-checkpoint gap in `w/o tool` accuracy between with-tool outcome-only RL and no-tool RL over all 21 evaluation checkpoints:
>
> | Backbone | VStar | HR4K | HR8K | VP(Easy) | VP(Medium) | VP(Hard) | Agg |
> | --- | ---: | ---: | ---: | ---: | ---: | ---: | ---: |
> | Qwen2.5-VL | 0.0127 | 0.0078 | 0.0144 | 0.0243 | 0.0179 | 0.0278 | 0.0175 |
> | Qwen3-VL | 0.0130 | 0.0123 | 0.0112 | 0.0284 | 0.0229 | 0.0269 | 0.0191 |
>
> HR = HR-Bench; VP = VisualProbe; Agg = aggregate over all six benchmarks.
>
> These results support using `Acc_wo` as a practical tool-free reference rather than as a perfectly isolated intrinsic measure.
>
> ## W3&Q3: Have you tried stronger interventions, such as reward shaping or forced call/no-call evaluations, to more directly isolate tool-learning effects?
>
> A: Beyond the hard test-time intervention already included in the paper via `Acc_schema` (Tab. 1 & 5), we additionally ran two training-time variants on both backbones for the same 200-step evaluation pipeline `[url, Fig.I]`: (i) no-tool RL, which removes tool access during RL training, and (ii) a tool-aware reward bonus (+0.1 for correct responses that call the tool).
>
> The test-time intervention shows that forcing schema presence while forbidding execution is not a better reference setting (-5.8 on Qwen2.5-VL; -13.0 on Qwen3-VL).
>
> The no-tool baseline shows that generic RL alone does not reproduce the same tool-related improvement: on Qwen2.5-VL it drives the model toward almost never calling (`P(c|D_fail)`: 0.2466 -> 0.0283; `P(c|D_succ)`: 0.3167 -> 0.0306), whereas with-tool outcome-only RL maintains active call-side behavior and a positive final `Gain - Harm`. The tool-aware reward variant shows that reward design matters, but mainly by changing call frequency and shifting the gain-harm tradeoff rather than by producing stronger selective tool use: on Qwen2.5-VL final `Gain - Harm` is 0.0134 under outcome-only reward vs. 0.0186 under tool-aware reward, whereas on Qwen3-VL it is 0.0582 vs. 0.0516.
>
> These interventions therefore make the diagnosis stronger within the current crop-and-zoom setting: generic RL alone does not recover the same tool-related dynamics, and a simple tool-call bonus mainly shifts call frequency and the gain-harm tradeoff rather than overturning our main finding that training reduces tool-induced harm more reliably than it improves correction on hard cases.
>
> url: https://anonymous.4open.science/r/anonymous-6124/
>
> [1] Pixel Reasoner. NeurIPS 2025
>
> [2] DeepEyesV2. ICLR 2026
>
> [3] Mini-o3. ICLR 2026
>
> [4] Thyme. ICLR 2026

---

> > ### Author Rebuttal · Reviewer_NwJo · 2026-04-02
> >
> > I am satisfied with the author's strong rebuttal. I plan to increase my score and hope that, if this paper gets accepted, it will incorporate the new discussion and experiment results.

---

> > > ### Author Response · Authors · 2026-04-02
> > >
> > > Thank you for your positive feedback on our work, particularly regarding the importance of the research question, the soundness of the Eq. 7 decomposition, the dynamic trajectory analysis, and our experimental design.
> > >
> > > We are thrilled that our rebuttal fully resolved your concerns and deeply appreciate your intention to raise your score. We will incorporate these valuable discussions and experimental results into the final version of the paper if accepted.
> > >
> > > Please feel free to let us know if you have any additional questions.

---

### Official Review · Reviewer_d1Zo · 2026-03-13

**Soundness:** 3
**Presentation:** 3
**Significance:** 2
**Originality:** 2
**Overall Recommendation:** 4
**Confidence:** 4

**Summary:**

This paper investigates the actual learning effect of vision tool-use reinforcement learning in vision-language models (VLMs), aiming to answer a central question: do the performance gains brought by tool-augmented RL come from improved intrinsic model capability, or from improved tool-use ability?

To study this, the authors propose an analysis framework called MED (Measure–Explain–Diagnose). First, by evaluating tool-free accuracy and tool-available accuracy at different training checkpoints, the overall performance change is decomposed into intrinsic drift and tool-induced drift. Next, the performance difference attributable to tools is further decomposed into four components: Call Gain, Schema Gain, Call Harm, and Schema Harm, representing positive or negative effects caused by tool invocation and tool schema. Finally, each component is further decomposed into Mass, Policy, and Quality factors to analyze changes in tool-calling strategies and execution quality.

Experiments are conducted on two VLMs (Qwen2.5-VL and Qwen3-VL) trained with GRPO, and evaluated on six benchmarks. The results suggest that most of the performance improvement comes from intrinsic capability gains, rather than improved tool usage. Moreover, RL mainly works by reducing the negative effects of tools (e.g., incorrect tool calls or schema interference), rather than improving the model’s ability to correct mistakes through tools. Based on these findings, the authors argue that current vision tool-use RL primarily learns to avoid tool-induced harm and coexist safely with tools, instead of truly mastering tool-use capability.

**Compliance With Llm Reviewing Policy:**

Affirmed.

**Final Justification:**

I acknowledge the value of this work, especially the two concrete implications for future tool-use RL, and therefore I am increasing my final score to Weak Accept.

However, I still view the analysis being limited to a single vision tool as a weakness of the work. An evaluation over a broader range of tools would, in my view, make the paper more clearly above the acceptance threshold.

**Key Questions For Authors:**

Q1. Generalizability of the findings. The analysis in this paper is mainly based on the crop-and-zoom tool. Have the authors attempted similar analyses on other types of tools (e.g., retrieval, OCR, or code execution)? Do the conclusions generalize more broadly?

Q2. Impact of the RL algorithm. The experiments use GRPO for training. Do the authors observe similar phenomena when using other RL methods (e.g., PPO or other RLHF variants)?

Q3. Variance and stochastic effects in evaluation. For a given checkpoint, repeated inference on the same instance may yield different outcomes due to sampling stochasticity. When analyzing the changes introduced by RL training, did the authors account for this variability? In addition, most analyses in the paper focus on macro-level metrics (e.g., dataset-level accuracy). Have the authors considered performing a more fine-grained analysis at the instance level, for example by examining the variance across multiple runs or the stability of tool-use behavior on individual examples?

**Limitations:**

yes

**Strengths And Weaknesses:**

## Strengths

1. A structured and systematic analysis framework. The proposed MED framework performs a multi-level decomposition (drift → gain/harm → mass/policy/quality) to analyze how tool usage affects performance changes. This provides a relatively clear analytical perspective for understanding the training dynamics of tool-use RL. The paper conducts checkpoint-level analysis during training and evaluates across multiple benchmarks, while also comparing tool familiarity across different models. Overall, the empirical analysis is relatively thorough.

2. A useful and practically relevant empirical observation. The paper attempts to disentangle intrinsic improvement from tool-induced improvement, and empirically finds that the main effect of tool-use RL is to reduce negative effects caused by tool usage, rather than improving the model’s ability to leverage tools to fix errors. This observation provides useful insight into the current role of vision tool-use RL and may help guide the community toward more effective directions for tool-use learning.


## Weakness

1. Limited experimental scope. The study focuses on a single tool (crop-and-zoom), two models, and a fixed RL training setup. As suggested by the title, the paper seems to use crop-and-zoom as a case study to analyze the behavior of vision tool-use during RL training. However, vision tool-use includes many other types of tools, and it remains unclear whether the conclusions generalize to other tools (e.g., retrieval, OCR, or code execution) or to different RL methods.

2. The methodological is somewhat ad-hoc decomposition. The MED framework essentially performs a probabilistic decomposition of performance differences and can be viewed as an analysis methodology. The proposed Gain/Harm and Mass–Policy–Quality decompositions appear somewhat manually designed. Overall, the method is closer to descriptive analysis, and the level of technical novelty is relatively limited.

3. Although the paper presents several interesting empirical observations, the main conclusion (that RL primarily reduces tool-related harm) is conceptually somewhat intuitive. Its direct implications for designing or improving tool-use RL algorithms remain limited.

---

> ### Author Rebuttal · Authors · 2026-03-31
>
> ##  W1&Q1: The study uses a single tool and a GRPO-based setup, so it is unclear how broadly the findings generalize across tools and RL methods.
>
>
> A: We appreciate this concern, and will clarify the intended empirical scope more explicitly: our evidence is best interpreted within the crop-and-zoom, GRPO/RLVR setting studied here. Crop-and-zoom is one of the most widely used visual tools in recent vision tool-use RL [1-4], so it is a well-motivated choice, and our benchmarks are standard in this setting. MED itself is more general: it only requires paired tool-free/tool-available evaluation plus observable call/no-call behavior. Here we instantiate it on a representative crop-and-zoom testbed with two backbones, six benchmarks, and checkpoint-level analyses.
> We further add no-tool RL and a tool-aware reward bonus `[url, Fig.I]` as nearby training-side checks within the same testbed, whose implications we discuss below.
>
> ## W2: MED may look somewhat ad-hoc and manually designed, so it is unclear whether the decomposition is principled or mainly descriptive.
>
> A: We agree that MED is an analysis methodology, but we do not think it is ad-hoc. Its novelty is analytical: it turns a single end-to-end gap into a diagnosis of where tool-use RL gains come from, beyond the usual reporting lens in prior work. The first decomposition starts from the paired gap `G(t)=Acc_w(t)-Acc_wo(t)` and partitions it by whether a sample is a tool-free failure/success (`D_fail` / `D_succ`) and by call / no-call, yielding the four Gain/Harm terms. Each of these four Gain/Harm terms is defined by a domain `D`, action `a`, and outcome `o`, and the later factorization is just the exact identity `Term(D,a,o)=P(D)P(a|D)P(o|a,D)`, i.e., how large the relevant failure/success subset is, how often the model makes that tool-call/no-call choice, and how often it succeeds or fails. So MED is descriptive by design, but not manually designed, and these decompositions enable Exp-II and Exp-III. We will revise the main text to make this two-step logic clearer.
>
> ## W3&Q2: The main conclusion may seem somewhat intuitive, and its direct implications for improving tool-use RL algorithms remain limited.
>
> A: We would respectfully disagree that this conclusion was already clear before this analysis. Prior work typically reports end-to-end tool-available performance, which does not separate stronger tool-based correction, broader tool-free improvement, and less tool-induced harm. MED makes these sources explicit, and Exp-I--III show that harm reduction dominates because hard-case correction remains limited while harm-side conditional errors drop more reliably. If the conclusion now seems intuitive, it is largely because the paper makes these sources measurable.
>
> This diagnosis yields two concrete implications for future tool-use RL.
>
> First, future methods should be evaluated by separating tool-free improvement from tool-interaction effects, not only by final tool-available accuracy. Exp-I--III show that end-to-end gains can come from broader tool-free improvement and reduced tool-induced harm, not only stronger tool-based correction. Our no-tool RL baseline `[url, Fig.II]` further shows that much of the tool-free improvement can be recovered without tool exposure during RL.
>
> Second, simply encouraging more tool use is not enough; future methods should target selective correction on `D_fail`, not raw call frequency. Our tool-aware reward variant `[url, Fig.I]` changes call frequency and the gain-harm tradeoff more than hard-case correction: on Qwen2.5-VL final `Gain - Harm` is 0.0134 under outcome-only reward vs. 0.0186 under tool-aware reward, while on Qwen3-VL it is 0.0582 vs. 0.0516. A future direction is a GRPO-style variant that performs paired tool-free/tool-available rollouts within each sample group and uses MED-inspired reward shaping to encourage correction on `D_fail` while discouraging Term-3/4 harms on `D_succ`.
>
> ## Q3: Did the analyses account for inference variability, and can the paper provide finer-grained evidence beyond macro-level averages?
>
> A: Greedy decoding with temperature=0 (Sec 4.1 & Appx.B.7) already reduces the usual inference-stochasticity concern in both settings. We also reran part of the evaluation for both backbones on three different machines; all observed stds stay below `0.011` in both tool-free and tool-available accuracy `[url, Tab.I]`.
>
> For aggregated metrics, Appx. G reports paired-bootstrap 95% confidence intervals. For finer-grained evidence, the paper tracks tool-based correction on the fixed failure set and persistent failures, and rebuttal adds a small instance-level case study of persistent failures `[url, Fig.III]`. Together, these analyses go beyond macro-level averages, and the repeated evaluations indicate small inference variability.
>
> url: https://anonymous.4open.science/r/anonymous-6124/
>
> [1] Pixel Reasoner. NeurIPS 2025
>
> [2] DeepEyesV2. ICLR 2026
>
> [3] Mini-o3. ICLR 2026
>
> [4] Thyme. ICLR 2026

---

> > ### Author Rebuttal · Reviewer_d1Zo · 2026-04-03
> >
> > I acknowledge the value of the two concrete implications for future tool-use RL, and therefore I am increasing my score to Weak Accept.
> >
> > However, I still view the analysis being limited to a single vision tool as a weakness of the work. An evaluation over a broader range of tools would, in my view, make the paper more clearly above the acceptance threshold.

---

> > > ### Author Response · Authors · 2026-04-03
> > >
> > > Thank you for the follow-up, and for recognizing both MED as a structured analysis framework and the value of the two concrete implications clarified in rebuttal.
> > >
> > > We agree that evaluating a broader range of vision tools would make the paper stronger, and we will make this empirical scope limitation even clearer in the paper. Our goal here is a focused study of vision tool-use RL in the crop-and-zoom setting, which is one of the most common and practically important visual operators in recent vision tool-use RL work.
> > >
> > > Within that scope, we hope the paper still makes a useful contribution by turning end-to-end gains into a concrete diagnosis of tool-free improvement, hard-case correction, and tool-induced harm. We will make this scope boundary clearer in the paper, while positioning broader validation across genuinely different vision tools as important future work.

---

### Decision · Program_Chairs · 2026-04-30

**Decision:**

Accept (regular)

**Comment:**

This paper studies an important question for VLM: do performance gains from tool-augmented RL arise from improved intrinsic model capability, or from better tool use? Through an analysis framework called MED (Measure–Explain–Diagnose) proposed by the authors, they conducted experiments on two VLMs (Qwen2.5-VL and Qwen3-VL) trained with GRPO across six benchmarks show that most performance improvements stem from enhanced intrinsic capabilities rather than improved tool use and suggest that current vision tool-use RL mainly helps models avoid tool-induced harm and coexist more safely with tools, rather than truly mastering tool-use capabilities.

The paper is well written, easy to understand and technically sound.

Therefore we recommend accept the paper.